# Number Cookbook: Number Understanding of Language Models and How to Improve It

**Haotong Yang**[123]   **Yi Hu**[12]   **Shijia Kang**[12]   **Zhouchen Lin**[1234*]   **Muhan Zhang**[23*]

[1] School of Intelligence Science and Technology, Peking University

[2] Institution for Artificial Intelligence, Peking University

[3] State Key Lab of General Artificial Intelligence, Peking University

[4] Pazhou Laboratory (Huangpu), Guangzhou, Guangdong, China

`haotongyang@pku.edu.cn  {huyi2002,kangshijia}@stu.pku.edu.cn`
`{zlin,muhan}@pku.edu.cn`

## Abstract

Large language models (LLMs) can solve an increasing number of complex reasoning tasks while making surprising mistakes in basic numerical understanding and processing (such as $9.11 > 9.9$). The latter ability is essential for tackling complex arithmetic and mathematical problems and serves as a foundation for most reasoning tasks, but previous work paid little attention to it or only discussed several restricted tasks (like integer addition). In this paper, we comprehensively investigate the **numerical understanding and processing ability** (NUPA) of LLMs. Firstly, we introduce a benchmark covering four common numerical representations and 17 distinct numerical tasks in four major categories, resulting in 41 meaningful combinations in total. These tasks are derived from primary and secondary education curricula, encompassing nearly all everyday numerical understanding and processing scenarios, and the rules of these tasks are very simple and clear. Through the benchmark, we find that current LLMs fail frequently in many of the tasks. To study the problem, we train small models with existing and potential techniques for enhancing NUPA (such as tokenizers, positional encoding, and number formats), comprehensively evaluating their effectiveness using our testbed. We also finetune practical-scale LLMs on our proposed NUPA tasks and find that 1) naive finetuning can significantly improve NUPA on many but not all tasks, and 2) surprisingly, techniques designed to enhance NUPA prove ineffective for finetuning pretrained models. We further explore the impact of chain-of-thought techniques on NUPA. Our work provides a more detailed and comprehensive understanding of NUPA in LLMs. Our benchmark and codes are released at https://github.com/GraphPKU/number_cookbook.

## 1 Introduction

The mathematical and reasoning abilities of large language models (LLMs) are currently quite impressive (OpenAI, 2023; Meta, 2024a; OpenAI, 2024a; Yang et al., 2024a), capable of solving problems at the level of a high-school student or even more difficult ones like GAOKAO (a nation-wide examination of high school students applying to universities in China) (Zhang et al., 2024b), Olympiad-level problems (He et al., 2024), and college mathematics (Tang et al., 2024). However, upon closer examination of the models' outputs, we find that although the models demonstrate remarkable proficiency in problem-solving approaches, they often struggle with basic numerical understanding and processing — like a careless student who claims, "*I know how to do it, but I didn't get it right.*" Some of these errors are quite surprising, such as believing that $9.11 > 9.9$ or making mistakes in simple addition $8/7 + 3/5$. These errors are a major cause of hallucinations when dealing with math, reasoning, and data analysis tasks, as the model presents seemingly correct problem-solving approaches, but ultimately produces incorrect results (Huang et al., 2024; Li et al., 2024b; Jiang et al., 2024). Therefore, investigating and improving the fundamental "**numerical understanding and processing abilities**" (NUPA) of models is crucial.

---

*Corresponding Authors

Table 1: Task overview of NUPA Test. The four rows represent four numerical representations, and the 17 columns correspond to different tasks. ✓: 41 tasks included in our test. ✗: Not included, too complex. ◯: Not directly included but can be easily adapted from an included task. −: Not applicable. The detailed explanation for these non-included tasks is provided in Appendix A.1.5

| | Elementary arithmetic | | | | | | Comparison | | Digit Understanding | | | | | | Conversion | | |
|---|---|---|---|---|---|---|---|---|---|---|---|---|---|---|---|---|---|
| | Add | Sub | Multiply | Truediv | Floordiv | Mod | Max | Min | Digit Max | Digit Min | Digit Add | Get Digit | Length | Count | To Float | To Scientific | Sig. Fig. |
| Integer | ✓ | ✓ | ✓ | ✓ | ✓ | ✓ | ✓ | ✓ | ✓ | ✓ | ✓ | ✓ | ✓ | ✓ | − | ✓ | ✓ |
| Float | ✓ | ✓ | ✓ | ✗ | − | − | ✓ | ✓ | ✓ | ✓ | ✓ | ✓ | ✓ | ◯ | − | ✓ | ✓ |
| Fraction | ✓ | ✓ | ✓ | ✓ | − | − | ✓ | ✓ | − | − | − | − | − | ◯ | ✓ | ◯ | ◯ |
| Scientific | ✓ | ✓ | ✓ | ✗ | − | − | ✓ | ✓ | − | − | − | − | − | ◯ | ✓ | − | ◯ |

However, in current research, **reasoning ability and NUPA are often tested together**, both on classic datasets such as GSM8k (Cobbe et al., 2021), MATH (Hendrycks et al., 2021b), MMLU (Hendrycks et al., 2021a), and in more challenging tests mentioned above. For example, a problem in GSM8k is: "*Natalia sold clips to 48 of her friends in April, and then she sold half as many clips in May. How many clips did Natalia sell altogether in April and May?*" Solving this problem requires two aspects: on the one hand, mathematical reasoning including understanding the text, extracting relevant information, formulating mathematical equations (or finding other solution methods), solving the equations or executing an algorithm, and obtaining the result; on the other hand, it also requires *understanding and processing the numbers* provided in the problem or produced as intermediate results at each step, like $48/2 = 24$ and $48 + 24 = 72$. While these two abilities are both essential to correctly solving the problems, tests on such datasets do not distinguish between them.

A more severe issue is that the numerical content is often **deliberately simplified** in these datasets. In various exam questions (like in the American Invitational Mathematics Examination (Li et al., 2024a)), to focus on assessing students' understanding of mathematical concepts — such as how to set up the correct equations and apply the right theorems — the numbers in both the questions and answers are often specially chosen to be **integers**. However, this is **not** the case in **real-world scenarios** (Chen et al., 2021).

Despite the importance of NUPA, there is still a lack of accurate, detailed, and comprehensive formalization, measurement, and analysis of this fundamental capability. In this paper, we take the preliminary step towards formalizing the NUPA of LLMs. We categorize the numerical concepts and operations from primary and secondary education into four representations: *integers*, *floating-point numbers* (finite decimals), *fractions*, and *scientific notation*, along with four ability categories comprising 17 tasks. Pairing these representations results in 41 meaningful tasks, forming our NUPA benchmark (Table 1). These representations and tasks cover the most common scenarios involving number understanding and processing, which are typically not challenging for humans, as we read, use, or process such numbers nearly every day.

On this benchmark, we rigorously test several state-of-the-art LLMs containing GPT-4o (OpenAI, 2024a), Llama-3.1 (Meta, 2024a) and Qwen2 (Qwen Team, 2024). We ask the models to directly output the answers without calling external tools. Although the latest LLMs perform well on some of the easiest tasks, their performance declines significantly as tasks become slightly more complex (such as multiplication, modulus operations, or digit-based calculations), or as the representation of numbers extends beyond basic integers. See Figure 2 of Section 2.4. The overall unsatisfactory performance highlights a pronounced mismatch between the claimed strong mathematical reasoning abilities and the poor *practical*, *everyday* numerical understanding and processing abilities of today's LLMs.

To address this issue, we explore three categories of approaches to enhance the NUPA of models. The first category of techniques aims at improving models' NUPA during the *pretraining* stage, including alternative tokenization, specially designed positional encoding (PE) (Haviv et al., 2022; Kazemnejad et al., 2023; Zhou et al., 2024b), changing number formats (like zero-padding, index-hint (Zhou et al., 2024a) and reverse representation (Lee et al., 2024; Zhou et al., 2024b)). We evaluate and analyze them on our newly introduced benchmark, verifying their effectiveness/ineffectiveness on respective tasks/representations, which extends over previous evaluation mainly on the integer addition/multiplication tasks. Further, we summarize these techniques into three mechanisms: simplifying the reasoning process, aiding digit alignment, and providing regularization, and discuss the potential of these mechanisms to be applied across a broader range of numerical representations.

The second category of approaches aim to improve NUPA for an *already trained* model. We find that while simple direct finetuning can significantly enhance NUPA performance, applying the

aforementioned techniques (PEs, data formats and tokenizers) at this stage may have *adverse effects*. We test various settings and finetuning configurations, but none are able to achieve performance equal to or better than the original model. Our results suggest that these modifications can significantly disrupt the models' established behavior or conflict with its pre-existing knowledge, leading to a decrease in performance.

Finally, we discuss the potential of using *chain-of-thought* (*CoT*) techniques (Wei et al., 2022) for numerical processing. Although CoT methods can break down complex problems into simpler sub-tasks and significantly increase the likelihood of obtaining correct answers, their drawbacks — such as consuming a large context window and requiring extended processing time — become particularly apparent in numerical tasks. We test a general CoT method known as RFFT (Hu et al., 2024), and find that for more complex tasks (such as multiplication and fraction addition), chain-of-thought methods face scalability challenges, making them difficult to be applied in practical scenarios. It is noteworthy that in this paper, we do not discuss tool use methods (Schick et al., 2023; Lu et al., 2023a) for NUPA as 1) we want to study the self-contained NUPA of LLMs, 2) calling external tools whenever encountering numbers increases the inference latency (Xu et al., 2024), and 3) we believe NUPA without tools is a necessary ability of AGI.

In summary, we propose a more comprehensive benchmark on the basic numerical understanding and processing abilities (NUPA) of LLMs, evaluate several SOTA LLMs' performance on it, and further study three categories of approaches to improve NUPA: pretraining, finetuning and CoT. Our results reveal that the current research is insufficient to fully address the NUPA problem, despite it being a fundamental capability for solving many more complex tasks. We hope that by introducing a systematic classification and more comprehensive evaluation of NUPA, we can bring greater attention from the community to this important but overlooked fundamental capability.

## 2 NUPA TEST: A BENCHMARK FOR NUMBER UNDERSTANDING AND PROCESSING ABILITY

In this section, we will introduce our NUPA benchmark from the following four aspects: number representations, tasks, metrics, and result analysis of current LLMs. We will explain the rationale behind the inclusion (or exclusion) of specific representations and tasks in our benchmark, highlighting their distinctive features.

### 2.1 NUMBER REPRESENTATION

As discussed above, we believe that the *educational curricula* on the (Chinese) primary and secondary school levels serve as a valuable reference for determining the essential NUPAs that LLMs should master. We identify four number formats in these curricula that are both common and sufficient to cover most practical scenarios.

- **Integer**: The most common number and the foundation of other number representations.
- **Floating-Point Number** (**Float**): Floats, or finite decimals, are a useful subset of fractions. Calculations with floats like addition and comparison, work similarly to integers, making them common in daily life.
- **Fraction**: We consider fractions with integer numerators and denominators. In practical situations involving distribution, fractions become unavoidable, especially when the inaccuracy introduced by converting fractions to floats is unacceptable.
- **Scientific Notation**: Scientific notation is characterized by separating a number's precise value from its order of magnitude. It is widely used in fields like physics, economics, and computer science because it efficiently handles a wide range of numbers and clearly conveys significant figures and precision. For LLMs, mastering scientific notation can significantly enhance their ability to handle practical tasks, such as interpreting financial reports or reading scientific texts.

Details of these four representations in our benchmark can be found in Appendix A.1.1. There are possible representations of numbers that are not included in these four formats, like *complex numbers*, *infinite decimal representation* (repeating and non-repeating), *radical expression* (like $\sqrt{2}$), ... These representations either occur infrequently in practical conversations (e.g., complex numbers) or present significant challenges for language models to process without the aid of external tools (e.g., radicals). For these reasons, we have opted not to include them in our benchmark at this stage.

## 2.2 TASKS IN FOUR ABILITY CATEGORIES

Another aspect of NUPA is defining the tasks that models need to handle. The tasks should have clear calculation rules. Furthermore, most practical numerical processing tasks should either fall within these tasks or can be easily transformed into some of them. Extracted from the primary and secondary education curricula, we propose 17 tasks in four ability categories and students who have completed the stage of education are expected to solve them. The complete task list is shown in Table 1 and we provide a more detailed discussion in Appendix A.1.2 and an example for each task in Appendix A.1.3. Below we discuss the rationales for including some tasks in detail.

- **Elementary arithmetic**: **addition**, **subtraction**, **multiplication**, and **division**. The most fundamental mathematical operations. For division, we consider three types of related operators: **True division, floor division** and **modulus**.

- **Comparison**: **max** and **min**. Understanding numbers on the concept of "order".

- **Digit understanding**: When we care about a language model's understanding, processing (and generation) of numbers, digit is a crucial concept, as numbers are not read and processed by the language model as a whole, but rather as a sequence of digits. We specially designed some digit-related tasks to test whether LLMs truly handle digits, including:
    - **Get digit**: Given a number and an integer $i$, return the $i$-th digit. This task is important when certain digits have special meanings in a number (such as a phone number or SSN).
    - **Length**: Return the total length (i.e., the number of digits) of a number.
    - **Count**: Count the times that a particular digit occurs in an integer.
    - **Digit compare**: Compare and return the larger (smaller) digits one by one.
    - **Digit add**: Perform the normal addition digit by digit but ignore any carrying. For example, $\mathrm{digit\_add}(12345, 34567) = 46802$. It can test a model's understanding of *digit alignment* and its mastery of single-digit addition.

- **Conversion between representations**: Converting a number to two representations: **to float** and **to scientific notation**, as they are frequently used to present final results. These two tasks test whether models can understand the relationship between various numerical formats. In particular, since many tasks present answers as approximate values, we designed a "**significant digit**" (**sig. fig.**) task to evaluate a model's ability to round long numbers to fixed-length significant digits.

The combination of representations and tasks ultimately results in a total of 41 meaningful pairs. Without confusion, we refer to each combination as a task. The tasks receive either one or two numbers as inputs and return a number as result, and the input numbers and results share the same representation for most tasks unless otherwise stated (refer to Appendix A.1.4). The remaining combinations are excluded due to being excessively complex, uncommon, inapplicable, or redundant with other tasks. For further details, see the discussion in Appendix A.1.5.

The difficulty of each task depends not only on the nature of the task itself but also on the *length of the numbers* to be processed — longer tasks involve longer inputs and outputs as well as more steps of internal operations. Therefore, we test on different problem lengths. For tasks that are inherently more difficult, we limit the size of the problem to 1-20 digits, and for easier tasks to 1-100 digits. (For which tasks are considered difficult or easy, please refer to the Appendix A.1.6.)

We generated 1,000 questions for each task and each length. Unlike some previous works that set the lengths of two numbers to be the same, in our tests, the length $L$ of a question is determined by the longer of the two numbers, while the length of the shorter number follows a uniform distribution between $L/2$ and $L$. We implemented additional handling to ensure that generated problems do not result in overly simple, complex, or meaningless results. Some tasks are further split into a hard and an easy version. More details about generating the benchmark are provided in Appendix A.1.7.

## 2.3 METRICS ABOUT NUPA

Measuring the performance of NUPA benchmarks on these tasks is not trivial. "**Exact match**" accuracy is the golden standard of the performance where the answer is considered as correct when it exactly matches the groundtruth. However, a smoother and more detailed metric is useful to understand the behavior and capabilities of a model. Therefore, we also report

| | |
|---|---|
| Generation: | 425.925535321 |
| Groundtruth: | 31415.92653582 |
| Exact match: | 0 |
| Digit match: | 8 / (8 + 5) =0.62 |
| dlength: | 3 |

Figure 1: An example of metrics.

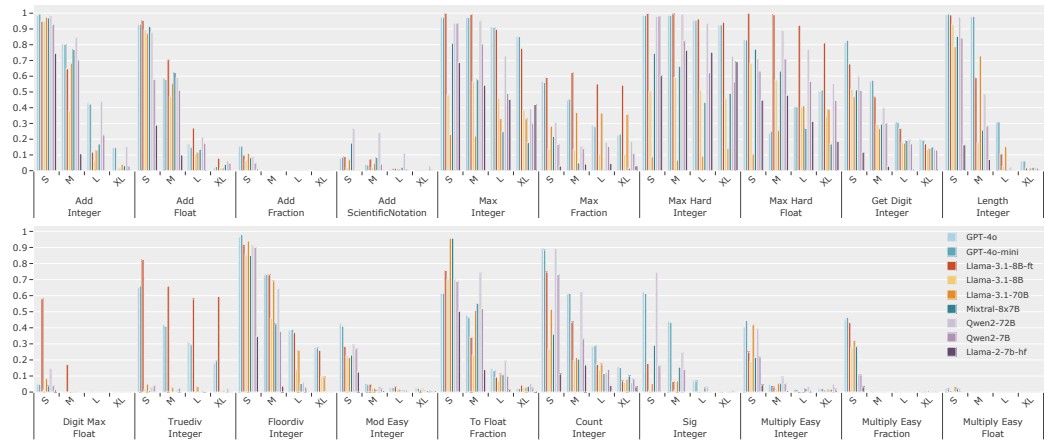

Figure 2: Parts of performance of state-of-the-art LLMs on NUPA benchmark.[1] "-ft" denotes a Llama model we finetuned on these tasks. (See Section 3.4)

the "**digit match**" and "**dlength**" (**difference of length**) metrics, as metrics of digit accuracy and length accuracy respectively. We first split numbers into parts (e.g., integer and decimal parts of a float, numerator and denominator of a fraction) and align the generated answer with the groundtruth digit by digit. Integer parts are aligned from the least significant digit; and the decimal parts of float are aligned from the most significant digit. For "digit match", we measure the correctness of each digit, with missing digits considered as errors, and report the overall accuracy. For "dlength", we report the sum of *absolute* difference in length between each part of the prediction and the groundtruth. Figure 1 illustrates these three metrics.

For each task, we divide the digits into four intervals (S, M, L, XL). For tasks with lengths 1-20, the four intervals correspond to 1-4 5-8, 9-14, 15-20 digits respectively. For tasks with lengths 1-100, they correspond to 1-10, 11-20, 21-60, 61-100 digits respectively. We average the results in each interval for each task and metric. More details of our metrics are given in Appendix A.1.8

## 2.4 PERFORMANCE OF CURRENT LLMS

We test some commonly used LLMs on our benchmark, including three Llama models: Llama-2-7b, Llama-3.1-8b and Llama-3.1-70b (Meta, 2024a), one of the most popular open-source model families from Meta; Mixtral-8×7B (MistralAI, 2024), a strong MoE model; and Qwen2-7B and Qwen2-72B (Qwen Team, 2024) which are also open-source models that are believed to have strong math abilities. Finally, we also test state-of-the-art commercial models GPT-4o-2024-08-06 and GPT-4o-mini-2024-07-18 (OpenAI, 2024a). We use prompts to control models to directly output result numbers without relying on external tools or CoT. The prompts used for each model and task are included in Appendix A.2. We select the results of some typical tasks in each category in Figure 2, while the complete results[1] and discussion on all metrics are shown in Appendix A.3. Here, we mainly focus on the zero-shot performance while we discuss few-shot performance in Appendix A.3.1. We have several observations regarding the results:

**The best model performs well on typical tasks, but its performance declines on more specialized tasks.** We find that GPT-4o, GPT-4o-mini and Qwen2 handle typical tasks, such as integer addition, float addition, integer max, and integer length, with high accuracy in the S and M ranges. This aligns with their strong performance on various mathematical datasets. However, their accuracy drops sharply when working with less common representations, like fractions and scientific notation, with average accuracy falling below 20%, even for the shortest S-range (1-4 digits). Similarly, for tasks such as significant figures, modulus operations, and digit-based calculations, their performance was unsatisfactory. This highlights the current limitations of LLMs in understanding numerical diversity and complexity. Despite their good performance on a narrow set of numerical tasks, they struggle with many others, failing to produce accurate results in these areas.

**Length remains a significant challenge for NUPA of LLMs.** We observe a noticeable decline in accuracy for even simple tasks like integer addition as the problem length increases. For instance, GPT-4o's accuracy drops from nearly 100% in the S range and 80% in the M range to around 40% in the L range and just 15% in the XL range. In the more complex task float addition, the accuracy

---

[1]An interactive performance report is shown in https://huggingface.co/spaces/kangshijia/NUPA-Performance.

decreases from 90% (S) and 60% (M) to merely 15% (L) and less than 5% (XL). This trend is consistent across other models and tasks. For example, Qwen2's performance in the integer-length task declines from almost 100% in the S range to 50% in the M range, and falls below 5% in the L and XL ranges.

**Length impedes learning both individual digits and overall length.** To understand why models struggle with longer input numbers, we examine *digit match* and *dlength* performance in Figure 6 and Figure 7 in Appendix A.3. These metrics reveal that length affects **both** the accuracy of individual digits (digit match) and the answer's overall length (dlength), with variations across tasks. For example, GPT-4o and Llama-3.1 display consistently low dlength in the add-integer task, with digit match decreasing sharply as length increases, suggesting that length primarily impacts per-digit accuracy on this task. Conversely, in the max-float task, dlength increases significantly with length (about 30-60 in the XL range), while digit match remains at 60% in the XL range. Note that since missing digits are treated as errors, this 0.6 digit match is likely due to these missing digits. This suggests that the main challenge here lies in generating answers of the correct length, rather than individual digit accuracy. In other tasks like fraction, both length and digit accuracy issues arise, as reflected in rising dlength and declining digit match.

**"Digit" is more challenging than expected.** We were surprised to find that LLMs struggle to fully grasp "digits". For instance, in the "get digit" task, where the model is asked to return the $i$-th digit of a long integer, performance drops significantly as the length of the number increases. This suggests that current LLMs lack a consistent ability to simply find a digit. Note that the performance is good in the shorter S-range, which indicates that the models can at least comprehend the task instruction. In the XL-range, GPT-4o achieves only 20% accuracy, barely above the random guessing 10% baseline (since the correct answer is always a digit between 0 and 9). This fundamental limitation may explain why current LLMs struggle with numerical understanding and processing, especially as task complexity and input length increase. If a model cannot reliably identify a specific digit in a given number, it casts doubt on its ability to generalize to more complex arithmetic tasks, such as addition.

We also have some interesting observations: (1) LLMs find the "max-hard" task easier than "max" with integer inputs. The difference between the tasks is that in the max task, the two numbers often differ in length, whereas in max-hard, they are always the same length and share some left-most digits, requiring more digits to be compared. While max-hard intuitively seems more difficult, models actually perform better on it. This is likely because they struggle to effectively use sequence length information, as reflected in their weaker performance on the "length" tasks in the longer ranges. It suggests that models might process tasks in different ways from humans. They could have to compare two numbers digit by digit. In this situation, the "harder" subtasks are actually easier because the numbers are already aligned. (2) GPT-4o and GPT-4o-mini show nearly identical performance across most tasks, similar to the comparison between Qwen2-72B and Qwen2-7B. This suggests that once a model reaches a certain size, NUPA performance relies more on factors like architecture, training strategies, data diversity, and post-training refinements, rather than simply on increasing model size.

## 3    HOW DO TOKENIZERS, PES AND DATA FORMATS AFFECT NUPA?

We have observed that the NUPA Test poses significant challenges even for the most advanced LLMs. In this section, we aim to investigate the factors that can influence the NUPA of LLMs during their pretraining phase, including tokenization strategies, PEs, and different data formats. We utilize the architecture of decoder-only transformers and alter the size to create models with 0.1B, 0.9B and 3B parameters. These models are trained from scratch, incorporating a wide range of techniques that could potentially impact NUPA. In this section, each model is trained on a *single* task . The details of the training process and models are included in Appendix A.4.1.

### 3.1    ONE-DIGIT TOKENIZERS ARE GOOD ENOUGH

LLMs interpret numbers as segmented tokens rather than whole numbers. With the development of language models, various tokenization strategies have emerged, including *mixed tokenizers*, *one-digit tokenizers*, and *k-digit tokenizers* ($k \geq 2$), as shown in Figure 3. In the BPE tokenizer used by GPT-2, the numbers are not specially treated, which resulted in irregular number cutting and is harmful to digit alignment. The cutting

(a) 31415.926535897932

(b) 31415.926535897932

(c) 31415.926535897932

(d) 31415.926535897932

Figure 3: Different tokenization of a long number. (a) GPT2: mixed digit tokenizer. (b) Llama-2: one-digit tokenizer. (c) GPT-3.5, GPT-4 and Llama-3: three-digit tokenizer. (d) Aligned three-digit tokenizer.

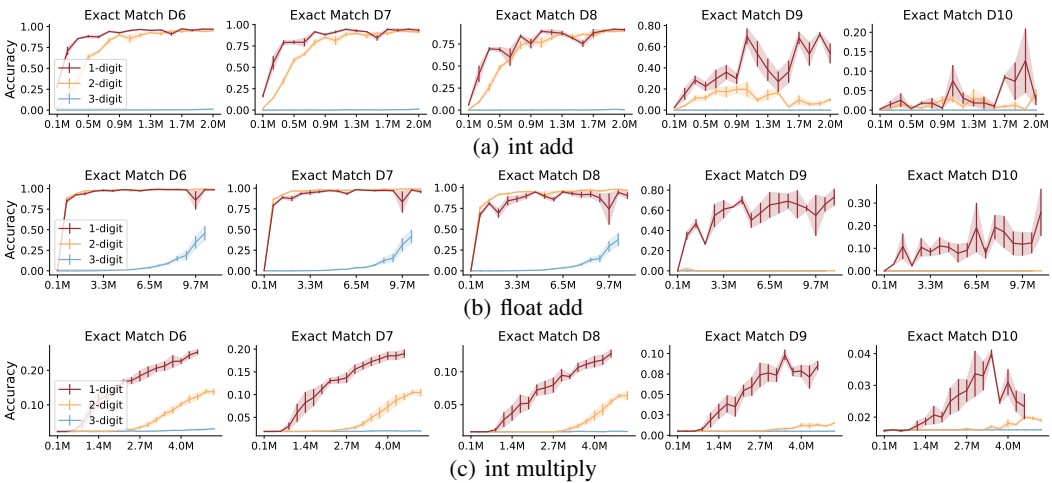

Figure 4: Accuracy of 0.9B models trained with 1-3 digit tokenizer on three tasks of integer addition, float addition and integer multiplication. Shadow shows the standard error. D$n$ means $n$ digits. X-axis is the number of seen training samples.

of numbers in modern tokenizers has become more aligned. These tokenizers greedily segment a number from left to right into $k$-digit tokens until a remainder shorter than $k$ digits is left, which is then segmented into a single token. Llama-2 uses a one-digit tokenizer, but all of the latest LLMs use a tokenizer with $k = 3$, which comes with an extended vocabulary for numbers. Additionally, Singh & Strouse (2024) discovers that just alternating the greedy direction from "left-to-right" to "right-to-left" (for integers) can improve performance of Llama-3 and GPT-4.

There is a growing tendency to expand the vocabulary size as the number of parameters in LLMs rapidly increases. Recent work has shown that a larger vocabulary is more suitable for larger LLMs (Tao et al., 2025) because longer tokens can encapsulate more complex and precise meanings for text tokens. However, numbers behave differently:

- The long-tail phenomenon (Raunak et al., 2020), common in text tokens, is not as pronounced for the number tokens. The distribution of number tokens is closer to a *uniform* distribution.
- Two smaller number tokens can always be combined into a valid new one (e.g., 3 and 7 form 37), which is not true for text tokens (e.g., "hello" and "hi" cannot form "hellohi"). So the number of possible number tokens grows exponentially as $k$ increases, much faster than text tokens.
- The next token prediction of number tokens is harder than predicting the next text token because number prediction often involves calculation and operations, whereas word mapping tends to be more intuitive.

We trained 0.9B models on 1- to 8-digit length samples including integer addition, float addition, and integer multiplication, using **aligned** $k$-digit tokenizers where $k = 1, 2, 3$ (d in Figure 3). Figure 4 shows the in-domain performance of these models in the first three columns and their out-of-domain (OOD) performance in the last two columns, evaluated using the exact match metric.

From the figure, the one-digit tokenizer shows the best in-domain performance in these three tasks, while three-digit tokenizer exhibits poor performance. In out-of-domain tests, the one-digit tokenizer also exceeds the others by large margins. Tokenizers with an increasing number of digits significantly hinder sub-billion models' NUPA. We also performed experiments on models of 3 different sizes including 0.1B, 0.9B, and 3B in Appendix A.4.2 and got similar results. Even as the model size increases, the performance of 2- or 3- digit tokenizer improves but remains either similar or worse than that of the one-digit tokenizers. For these experiments, we also report the digit match and dlength results in Appendix A.4.2 Figure 10 and 11, where one-digit tokenizer performs better both on digit learning (larger digit match) and length learning (less dlength). On the contrary, larger vocabularies significantly increase the model size requirements. In conclusion, we find **no evidence** to support the idea that increasing the vocabulary size improves NUPA performance.

Recently, Sathe et al. (2024) found that the "**random tokenizer**" (Kudo, 2018; Provilkov et al., 2020) which splits words like "Hello world" into variable tokens such as "He/llo/ world" or "Hell/o/ world" enhances reasoning by introducing variability in generation path. We also test it in number domain and find the random tokenizers consistently outperform their standard counterparts in length generalization, but **still fall short** of the performance achieved by the one-digit tokenizer. See the details in Appendix A.4.2.

## 3.2 SPECIALIZED PEs ACT AS LENGTH REGULARIZERS

Previous work has suggested that PE could be the key factor (Zhou et al., 2024b) of length generalization. To further investigate whether the influence is specific on a certain task, we train 100M models with different PEs: RoPE (Su et al., 2024), NoPE (Kazemnejad et al., 2023) and Alibi (Press et al., 2022) on four tasks: *integer addition*, *float addition*, *fraction multiplication (easy)* and *scientific notation addition* respectively. Models are trained on 1-8 lengths (S and M range), then test them on full range (S to XL, 1-20). RoPE, widely used in Llama and its derivatives, is the most classic relative PE. Then Alibi, another relative PE, is proposed to address RoPE's length overfitting issues. NoPE (transformers without PE, relying solely on the causal mask to encode the position information) offers a surprisingly easy way to achieve length generalization. Therefore, we compare these three typical PEs to evaluate the performance on NUPA.

Our results, presented in Figure 13 in Appendix A.4.3, align with conclusions from previous works. Alibi and NoPE demonstrate superior length generalization across various representations and tasks, indicating that the influence of PEs is relatively consistent across these common representations, tasks within the number domain.

Moreover, we aim to characterize further the mechanism underlying these differences. Specifically, we found that RoPE leads the model to learn a length-related *shortcut*, while Alibi and NoPE act as a form of *regularization* by avoiding this, thereby preventing length overfitting. For more details, please refer to the appendix A.4.3.

## 3.3 DATA FORMATS HELP DIGIT ALIGNMENT

A series of works have proposed specific data formats including reverse formatting, zero padding and index hints. **Reverse formatting** (Lee et al., 2024; Shen et al., 2023) presents numbers in reverse order from the least significant digit to the most significant one to align with the models' autoregressive mechanism, simplifying the learning process for addition. **Zero padding** (Lee et al., 2024; Shen et al., 2023; Zhou et al., 2024b; Cho et al., 2024) adds leading zeros to numbers to standardize the lengths of operands, helping models align operands. **Index Hints** (Zhou et al., 2024a) explicitly incorporate positional information by prefixing each digit with its corresponding position index in both input and output sequences.

While previous work mainly focuses on integer addition or multiplication, we extend the techniques to various tasks in the NUPA Test of different number domains. To compare the effects of reverse formatting and zero padding, we demonstrate in Table 16 how the combination of reverse formatting and zero padding impacts length generalization. **Reverse formatting, zero padding, and their combination all outperform** vanilla formats in integer and float addition, while their performance is comparable to each other, suggesting that their functionality largely **overlaps**. Zero padding helps ensure proper alignment, while reverse formatting also plays a crucial role in maintaining alignment. The previously believed "*helping calculation*" function of reverse formatting is *minor*. As for index hint, we find it doesn't work for our models. We discuss the details of these experiment results and the reasons in Appendix A.4.4.

## 3.4 DOES FINETUNING IMPROVE NUPA PERFORMANCE OF LLMS?

The existing techniques aimed at enhancing NUPA have rarely been applied to practical LLMs, mostly being tested on toy models and isolated tasks. This raises the question of whether it is possible to enhance the NUPA capabilities of large models through post-training finetuning. To explore this, we generate training sets ($10^5$ samples for each digit and each task) and validation sets for our NUPA tasks, ensuring no overlap with the original test set. We then use them to finetune a pretrained model. Specifically, we finetune a Meta-Llama-3.1-8B model with LoRA (Hu et al., 2022) (rank 128, $\alpha$=32) on a **mixed** training set comprising all of our NUPA tasks. Remarkably, we find only 800 steps training (about 50M training samples, $\ll$ 1 epoch) leads to significant improvement, as shown in Figure 2 with the finetuned model labeled as "`Llama-8B-ft`". Though Llama-3.1-8B is not a strong baseline, this **finetuned version achieves much better performance**. For example, in max, max-hard, add-float and truediv tasks, this model even surpassed or matched GPT-4o, confirming our hypothesis: for many NUPA tasks, the model's base capacity may not be the main limiting factor, but rather *the lack of numerical diversity* and *task variety* in the training data.

However, we also found that such finetuning does not provide much improvement on certain tasks, such as understanding digits. Furthermore, when we tried to incorporate the various tricks, such as modifying the model's original PEs, tokenizers, or number formats, into an *already trained* model, these methods proved **ineffective**. When we altered the PE or adjusted the tokenization and representation of the model, the changes significantly disrupted the model's original behavior, causing a substantial performance drop. This suggests that enhancing a model's NUPA capabilities through post-training may require more revolutionary innovations beyond the current tricks. The detailed results of these attempts are presented in Table 18 in Appendix A.4.5.

## 4 IS COT SUITABLE AND VALID FOR NUPA?

CoT has been proven to be effective in enhancing the capacity of LLMs both theoretically (Feng et al., 2023; Yang et al., 2024b) and experimentally (Wei et al., 2022; OpenAI, 2024b). Thus, we are also interested in whether CoT is the ultimate solution for improving NUPA. Due to the task and representation diversity in our benchmark, it is hard to cover all issues with a single form of CoT. So we adapt a special CoT form called Rule-Following CoT (Hu et al., 2024) (RF-CoT), where LLMs are trained to follow a provided code or pseudo-code that outlines the procedure to solve the task. RF-CoT is capable of handling any problem with a solving procedure that can be broken down into recurrences and basic unit operations, making it well-suited for our benchmark tasks. The detailed introduction with an example of RF-CoT can be found in Appendix A.5.1.

Table 3: Performance of RF CoT. "-" means exceeding context window limitation (2k tokens).

| Exact Match | Add Float | | | Multiply Fraction | | | Max Scientific | | | Mod Integer | | |
|---|---|---|---|---|---|---|---|---|---|---|---|---|
| # Digit | 5 | 6 | 7 | 2 | 3 | 4 | 38 | 39 | 40 | 6 | 7 | 8 |
| RF CoT | **1.00**±.00 | **1.00**±.00 | - | **0.93**±.01 | **0.88**±.03 | - | **1.00**±.00 | **1.00**±.00 | **1.00**±.00 | **0.67**±.05 | **0.43**±.07 | - |
| GPT-4o | 0.78 | 0.66 | 0.49 | 0.53 | 0.20 | 0.00 | 0.37 | 0.46 | 0.36 | 0.01 | 0.00 | 0.00 |
| Qwen2-72B | 0.62 | 0.50 | 0.70 | 0.05 | 0.00 | 0.00 | 0.96 | 0.98 | 0.95 | 0.03 | 0.00 | 0.00 |
| Llama-8B-ft | 0.88±.02 | 0.79±.04 | **0.74**±.04 | 0.50±.02 | 0.20±.03 | **0.01**±.00 | 0.98±.01 | 0.97±.01 | 0.98±.01 | 0.08±.02 | 0.05±.04 | **0.05**±.04 |

To evaluate the performance of this CoT method, we finetuned the LLaMA 3.1-8B model on a subset of the NUPA tasks with RF-CoT. During both training and testing, we set a context window of 2000 tokens, with any data exceeding this limit being ignored. Table 3 shows the performance on selected tasks. Accuracy and standard error for RF-CoT and finetuned

Table 2: Average inference time.

| | batchsize | sec / sample |
|---|---|---|
| RF CoT | 128 | 5.625 |
| Direct | 128 | 0.371 |
| Direct | 256 | 0.336 |

Llama-3.1-8B are averaged over three runs. For GPT-4o and Qwen2, which are not finetuned, we report single-run accuracy without standard error. Within the context length limit, the rule-following finetuned LLaMA 3.1-8B significantly **outperformed** GPT-4o and Qwen2-72B as well as the one finetuned without RF-CoT in most situations.

However, it requires a significantly *longer context window* and causes much *slower inference speed* compared to directly generating the answer. As shown in Table 3, with the 2000-token limit, CoT can only handle fraction addition involving numbers up to three digits. We provide the maximal digit length within the 2k context window limitation for each task in Appendix A.5.2 to show the context window limitation for complex tasks. As for inference time, Table 2 demonstrates the average inference time for generating each sample using "RF-CoT" and "direct answer" during the NUPA Test where both experiments are operated on an A800 GPU. In the table, the "direct answer" with batch size 256 uses a similar amount of CUDA memory as RF-CoT with batch size 128. The RF-CoT method is approximately 17 times slower than directly generating the answer, causing an unsustainable burden for such a basic operation that is frequently encountered in solving real-world problems, especially considering that number calculations may only account for a small part of a complex, real-world reasoning problem (such as analyzing a financial report).

## 5 RELATED WORK

We have discussed some related work in the corresponding section. This section highlights some other studies related to NUPA in language models.

**Numerical understanding in natural language comprehension** Earlier studies explored numerical reasoning within language comprehension contexts. For example, Dua et al. (2019) introduced a reading comprehension dataset requiring discrete reasoning, such as sorting and addition. Similarly,

Ravichander et al. (2019) proposed a benchmark for evaluating quantitative understanding in textual entailment. However, these datasets blend numerical reasoning with broader language understanding tasks, making it challenging to isolate numerical processing abilities.

**Probing numerical understanding in LMs**  Several works have probed numerical comprehension in encoder models. Wallace et al. (2019) trained probing models to assess numerical understanding embedded in model representations, while Johnson et al. (2020) extend this conclusion to multi-language settings. Naik et al. (2019) used contrastive tests to evaluate models' understanding of number magnitudes. Geva et al. (2020) demonstrated that finetuning on numerical reasoning data enhances the understanding. Unlike these studies, which focus on embeddings, our work emphasizes generating correct answers in autoregressive models. Recent efforts on such models include Razeghi et al. (2022), who studied few-shot learning correlations between term frequency and performance, and Zhang et al. (2024a), who identified key components in LLMs for basic arithmetic tasks. These works focus on some most classic tasks and our benchmark expands on these by incorporating diverse numerical representations, tasks, and digit ranges, offering a more comprehensive analysis.

**Numerical dataset in specific domains**  Datasets like those proposed by Spithourakis & Riedel (2018) and Lin et al. (2020) test numerical commonsense reasoning, while others focus on specific contexts, such as financial reasoning (Chen et al., 2021; 2022) or tabular data (Akhtar et al., 2023). These works highlight numerical reasoning within specific domains rather than general numerical processing tasks. In contrast, our benchmark targets core numerical understanding, emphasizing tasks decoupled from domain-specific constraints.

**Mathematical Reasoning Datasets**  Despite its close relationship with NUPA, *mathematical reasoning* is a broader field involving diverse skills such as task comprehension, equation solving, tool usage, and more (Lu et al., 2023b). While correct numerical processing is a critical component of mathematical reasoning, it is not the entirety of it (Stolfo et al., 2023). Datasets like MathQA (Amini et al., 2019), GSM8k (Cobbe et al., 2021), MATH (Hendrycks et al., 2021b), and SVAMP (Patel et al., 2021) focus on math word problems requiring multi-step reasoning and problem-solving. Few works isolate numerical processing from mathematical reasoning. Saxton et al. (2019) introduced a dataset for numerical tasks, such as adding floating-point numbers, but lacked task categorization by difficulty or length. Moreover, mixing numerical and algebraic tasks complicated analyses of pure numerical processing. Our benchmark addresses this gap, offering fine-grained categorization and evaluation of numerical understanding tasks.

## 6 CONCLUSION

We investigate NUPA of LLMs and introduce a comprehensive benchmark, the NUPA Test, to reveal that numerical problems remain challenging for modern LLMs. Our comprehensive test, which includes a variety of numerical representations and tasks, has exposed the surprising vulnerability of LLMs in this fundamental area. To explore ways to improve NUPA, we extend and evaluate previous pretraining techniques on the NUPA benchmark. While direct finetuning on the NUPA tasks does improve the performance, utilizing those tricks specifically designed for NUPA in the finetuning tends to harm NUPA, suggesting that these methods are not easily transferable to practical LLMs. We also explore the potential of chain-of-thought techniques to enhance NUPA and discuss their limitations.

## 7 LIMITATION

As a benchmark that specifically focuses on number understanding and processing abilities, we acknowledge that the range of tasks could still be incomplete and biased toward certain aspects. We will continue updating our benchmark, including but not limited to adding new tasks and refining existing ones to ensure appropriate difficulty. Additionally, the number of models we have tested so far is limited, and we plan to include more promising pretrained models in future evaluations.

On the other hand, although we have identified the limitations of LLMs' NUPA, the existing solutions each have their own drawbacks. We have yet to find a path that fully addresses the problem. Solving this issue may require research across multiple fields, such as enhancing the diversity of pretraining corpora, developing new techniques, or enabling more efficient reasoning paradigms that make more complex CoT approaches feasible. We hope our work can contribute to and be complemented by advancements in these areas.

## REPRODUCIBILITY STATEMENT

We have made every effort to ensure that the results presented in this paper are fully reproducible. Detailed descriptions of the number formats, construction and metrics of our NUPA dataset are provided in Section 2 and A.1.5, and examples for each task in A.4.4. To further facilitate reproducibility, we have included the complete dataset and the source code, enabling the generation of the entire dataset and the training and assessment of models, within the supplementary materials and the github page https://github.com/GraphPKU/number_cookbook. Researchers wishing to generate NUPA benchmark or replicate our experiments can refer to these resources for all necessary information.

## ETHICS STATEMENT

In conducting this research, we have adhered to the highest ethical standards to ensure the integrity and fairness of our work. For source code releases, we have ensured compliance with applicable legal standards. During the construction of the dataset, all data was entirely generated randomly, without including any personal identity information or other private data of individuals.

## ACKNOWLEDGEMENTS

This work was supported by National Key R&D Program of China (2022ZD0160300) and the NSF China (No. 62276004).

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

## A APPENDIX

### A.1 NUPA TEST

#### A.1.1 REPRESENTATIONS

We present the four representations as follows:

- **Integer**: we use no comma or point as a *digit group separator* like 1234567. The integer has only one part as itself. In this paper, we have not considered negative numbers for the time being.

- **Float**: A float has two parts: integer and decimal. We use a decimal point to split these two parts and also do not use any digit group separator. An example is 1234.567891. Trailing zeros in the decimal part are usually omitted.

- **Fraction**: A fraction has two parts: numerator and denominator and we use a "/" to separate the numerator and denominator parts. Unless otherwise specified, all fractions mentioned in this paper are in their simplest form (that is the numerator and denominator are coprime),

but they may be greater than 1. An example is 12/7. Only in the "truediv" task between two fractions, because the "/" is also the division operator, we enclose fractions in a pair of parentheses like (12/7) / (2/3) = 18/7 to make it clear.

- **Scientific Notation**: A scientific notation has two parts: significand and exponent. In our benchmark, the significand is always a float larger than 1 and less than 10 and the exponent should be a positive integer (and we also set an upper bound of 99). We use a "e" to separate these two parts. An example is 1.5e8.

### A.1.2 DETAILED INTRODUCTION AND DISCUSSION ABOUT TASKS

In addition to the brief introduction of the 17 tasks in our benchmark, here we provide a detailed discussion on why these tasks are significant and the specific abilities they aim to evaluate.

- **Elementary arithmetic**: **addition**, **subtraction**, **multiplication**, and **division**. They are the most fundamental mathematical operations and the first branch of mathematics taught in schools. However, some operations can be complicated when different number representations are involved. For example, fraction addition is more complicated than multiplication because it needs to be reduced to a common denominator first.
  - **True division, floor division and modulus**: The division is somewhat unique because it is not closed for integers and floats. Here, we consider three common division-related calculations. **True division**: To maintain precision, we represent the division of two integers as a simplified fraction. Combined with the "significant digits" task we will mention later, this can approximate the result of dividing two integers as a float. **Integer division** and **modulus**: Represent approximate multiple relationships, frequently used in practical applications, such as dividing individuals into batches.
- **Comparison**: **max** and **min**. Another important aspect of understanding numbers lies in the concept of "order". To truly comprehend a number, we must know how large it is and whether it is greater or smaller than another one. Moreover, comparison serves as the foundation for other significant operations. For instance, when adding negative and positive numbers, we determine the sign first and then subtract with their absolute values — this involves identifying which of the two numbers has a greater absolute value.
- **Digit understanding**: The concept of a digit is fundamental. Unlike the "value" of a number, a digit is tied to its specific representation. When we care about a language model's understanding, processing (and generation) of numbers, digit is a crucial concept, as numbers are not read and processed by the language model as a whole, but rather as a sequence of digits. We are curious whether LLMs truly understand the concept of digits. Therefore, we specially designed some digit-related tasks, including:
  - **Get digit**: Given a number and an integer $i$, return the $i$-th digit. This task is important when certain digits have special meanings in a number (such as a phone number or SSN).
  - **Length**: Return the total length (i.e., the number of digits) of a number.
  - **Count**: Count the times that a particular digit occurs in an integer.
  - **Digit compare**: Compare and return the larger (smaller) digits one by one.
  - **Digit add**: Perform the normal addition digit by digit but ignore any carrying. For example, $\text{digit\_add}(12345, 34567) = 46802$. It can test a model's understanding of *digit alignment* and its mastery of single-digit addition.

Through these tasks, we can assess whether models correctly understand the concepts of digits, length, positions, and the alignment of the digits between two numbers.

- **Conversion between representations**: we design tasks for converting a number to two representations: **to float** and **to scientific notation**, as they are frequently used to present final results. These two tasks also create transformations between different representations to test whether models can understand the relationship between various numerical formats. In particular, since many tasks present answers as approximate values, we designed a "**significant digit**" (**sig. fig.**) task to evaluate a model's ability to round long numbers to fixed-length significant digits.

### A.1.3 EXAMPLES FOR EACH TASK

We provide each tasks with an example. To test the models, we also add some model specific system messages like "You are a helpful assistant to process numbers. Please directly answer the question

after the "=". The context before "=" is the question and the context after "=" is the groundtruth and is removed when testing.

- Add-Integer: Add two numbers: $744 + 543 = 1287$
- Add-Float: Add two numbers: $93.81 + 9.976 = 103.786$
- Add-Fraction: Add two numbers: $3/8 + 2/5 = 31/40$
- Add-Scientific: Add two numbers: $9.92e16 + 9.731e18 = 9.8302e18$
- Sub-Integer: Subtract two numbers: $744 - 543 = 201$
- Sub-Float: Subtract two numbers: $93.81 - 9.976 = 83.834$
- Sub-Fraction: Subtract two numbers: $2/5 - 3/8 = 1/40$
- Sub-Scientific: Subtract two numbers: $9.731e38 - 9.92e36 = 9.6318e38$
- Multiply-Integer: Multiply two numbers: $968 \times 8 = 7744$
- Multiply-Float: Multiply two numbers: $8.4 \times 9.555 = 80.262$
- Multiply-Fraction: Multiply two numbers: $8/7 \times 5/2 = 20/7$
- Multiply-Scientific: Multiply two numbers: $9.92e16 \times 9.731e38 = 9.653152e55$
- Truediv-Integer: Divide two numbers and return the result as a fraction. $744 / 543 = 248/181$
- Truediv-Fraction: Divide two numbers and return the result as a fraction. $(3/8) / (2/5) = 15/16$
- Floordiv-Integer: Divide two numbers and return the result as an integer. $845 // 152 = 5$
- Mod-Integer: Divide two numbers and return the remainder. $845 \% 152 = 85$
- Max-Integer: Get the maximal number: $50404$ and $97871 = 97871$
- Max-Float: Get the maximal number: $44.418$ and $65.669 = 65.669$
- Max-Fraction: Get the maximal number: $3/5$ and $3/8 = 3/5$
- Max-Scientific: Get the maximal number: $8.15e64$ and $1.063e73 = 1.063e73$
- Digit_max-Integer: Compare two numbers digit by digit and return the larger digit at each position, treating any missing digits as 0. $50194$ and $14283 = 54294$
- Digit_max-Float: Compare two numbers digit by digit and return the larger digit at each position, treating any missing digits as 0. $35.905$ and $8.4 = 38.905$
- Digit_add-Integer: The task is to add two given numbers digit by digit and return the result modulo 10 (ignoring carry), treating any missing digits as 0. $50404$ digit add $97871 = 47275$
- Digit_add-Float: The task is to add two given numbers digit by digit and return the result modulo 10 (ignoring carry), treating any missing digits as 0. $44.418$ digit add $65.669 = 9.077$
- Get_digit-Integer: Get the digit at the given position (from left to right, starting from 0). $50404$ at position $4 = 4$
- Get_digit-Float: Get the digit at the given position (from left to right, starting from 0). $44.418$ at position $3 = 1$
- Length-Integer: The total number of digits of $50404 = 5$
- Length-Float: The total number of digits of $262.534 = 6$
- Count-Integer: Count the number of the given digit in the given number: $27422$ count the occurrence time of digit $2 = 3$
- To_float-Fraction: Convert the number to float: $9/5 = 1.8$
- To_float-Scientific: Convert the number to float: $8.538e2 = 853.8$
- To_scientific-Integer: Convert the number to scientific notation: $50400 = 5.04e4$
- To_scientific-Float: Convert the number to scientific notation: $262.534 = 2.62534e2$

- Sig.Fig-Integer: Convert the number to scientific notation: $50194$ and keep significant figures as $3 = 5.02e4$
- Sig.Fig-Float: Convert the number to scientific notation: $65.669$ and keep significant figures as $2 = 6.6e1$

### A.1.4 EXPECTED REPRESENTATION IN EACH TASK

Each task in the 41 ones receives one or two input numbers and expects one number as the result. We name the representation by the first input numbers. For simplicity, the second input number shares the same representation as the first one for most tasks. Calculations between different representations can be performed by first converting them to the same representation. Two types of tasks are the exception. Tasks "length", "to float" and "to scientific" do not have the second input. The second inputs in tasks "get digit", "count", "sig. fig." are always a short Integer, representing a position, length, or a digit number from 0 to 9. To distinguish them from potentially long integers to be processed, we call the former int and the latter integer.

We summarize the second number representation and result representation in each task in Table 4 and Table 5 where I means integer, i means (shorter) int, Fl means float, Fr means fraction, S means scientific notation and N means no such a number.

Table 4: The second input number representation

| | Elementary arithmetic | | | | | | Comparison | | Digit Understanding | | | | | | Conversion | | |
| | Add | Sub | Multiply | Truediv | Floordiv | Mod | Max | Min | Digit Max | Digit Min | Digit Add | Get Digit | Length | Count | To Float | To Scientific | Sig. Fig. |
| --- | --- | --- | --- | --- | --- | --- | --- | --- | --- | --- | --- | --- | --- | --- | --- | --- | --- |
| Integer | I | I | I | I | I | I | I | I | I | I | I | i | N | i | — | N | i |
| Float | Fl | Fl | Fl | ✗ | — | — | Fl | Fl | Fl | Fl | Fl | i | N | ○ | — | N | i |
| Fraction | Fr | Fr | Fr | Fr | — | — | Fr | Fr | — | — | — | — | — | ○ | N | ○ | ○ |
| Scientific | S | S | S | ✗ | — | — | S | S | — | — | — | — | — | ○ | N | — | ○ |

Table 5: Result number representation

| | Elementary arithmetic | | | | | | Comparison | | Digit Understanding | | | | | | Conversion | | |
| | Add | Sub | Multiply | Truediv | Floordiv | Mod | Max | Min | Digit Max | Digit Min | Digit Add | Get Digit | Length | Count | To Float | To Scientific | Sig. Fig. |
| --- | --- | --- | --- | --- | --- | --- | --- | --- | --- | --- | --- | --- | --- | --- | --- | --- | --- |
| Integer | I | I | I | Fr | I | I | I | I | I | I | I | i | i | i | — | S | S |
| Float | Fl | Fl | Fl | ✗ | — | — | Fl | Fl | Fl | Fl | Fl | i | i | ○ | — | S | S |
| Fraction | Fr | Fr | Fr | Fr | — | — | Fr | Fr | — | — | — | — | — | ○ | Fl | ○ | ○ |
| Scientific | S | S | S | ✗ | — | — | S | S | — | — | — | — | — | ○ | Fl | — | ○ |

### A.1.5 NON-INCLUDED TASKS

We exclude some compositions between number representations and tasks because of the following three reasons:

- ✗ too complex. We exclude the truediv between float and scientific. Division between float numbers is difficult to define accurately in our scenario. It is very common to divide two floating point numbers into an infinite decimal, which means that even very short decimals can still result in a very long and unpredictable result after division. And in this task we do not want to discuss the case of rounding the result. (This is another task of ours.) For the same reason, we also exclude division in scientific notation.

- ○: can be easily transferred to from an included task.
  - Converting fractions to scientific notation can be done by first converting to a float. (Fraction-to_scientific = Fraction-to_float + Float-to_scientific). Fraction-SignificantFigure is similar.
  - Scientific notation retains significant digits and is virtually identical to floating point numbers.

- count is a special task where we just consider a number as "a set of digits" so count in a float, fraction and scientific notation is as the same as in a integer.

- −: not applicable.

  - In fraction and scientific notation, the digit concept is not well-defined so the tasks about digit (digit-compare, digit-add, get-digit and length) are not applicable.
  - Floordiv and mod is only defined on integer.
  - Integer and float do not need to be further converted to float. Similarly, scientific has no need to converted to scientific.

### A.1.6 EASY/HARD SPLIT OF NUPA TASKS

We divide the tasks into easy and hard as shown in Table 6, where the hard tasks marked as H with maximal test digit as 20 and the easy tasks marked as E with maximal test digit as 100.

Table 6: Tasks can be divided into Easy and Hard.

| | Elementary arithmetic | | | | | | Comparison | | Digit Understanding | | | | | | Conversion | | |
|---|---|---|---|---|---|---|---|---|---|---|---|---|---|---|---|---|---|
| | Add | Sub | Multiply | Truediv | Floordiv | Mod | Max | Min | Digit Max | Digit Min | Digit Add | Get Digit | Length | Count | To Float | To Scientific | Sig. Fig. |
| Integer | H | H | H | H | H | H | E | E | E | E | E | E | E | E | | E | E |
| Float | H | H | H | | | | E | E | E | E | E | E | E | | | E | E |
| Fraction | H | H | H | H | | | H | H | | | | | | | H | | |
| Scientific | H | H | H | | | | E | E | | | | | | | E | | |

### A.1.7 PREPROCESS AND QUESTION GENERATION FOR NUPA TASKS

We define the length of a number as the number of digits in the *longest part* of a number. The "*integer*" part and "*decimal*" part of a float (as well as the significand of a scientific notation), the "*numerator*" and "*denominator*" of a fraction, the "*exponent*" of a scientific notation are considered as different "*parts*". In order to generate a pair of numbers with the larger length $L$, we first generate a $L$-length number and then generate a $l$-length number where $l$ follows a uniform distribution from $L/2$ to $L$. If the operation is commutative, we swap the two numbers with probability 0.5.

After we select two random numbers, we have some preprocessing to generate the final questions:

- For "Multiply", the difficulty also affected by the shorter number severely, so we split the task into two sub-tasks as "multiply-hard" and "multiply-easy". For the hard subset, we require that the shorter number must be longer than half of the longer one. For an easy subset, we require that the length of the shorter number is less than 3, so that the complexity is $O(n)$ instead of $O(n^2)$. And because the addition of fractions also involves multiplication, we also add an add-easy for this task in the same way.

- For "max" and "min" tasks, we additionally provide a harder version. For Integers and floats, we make sure that two compared numbers share the same length. At the same time, they should have more digits as the same like 12949 and 12961 to avoid models that can solve the problem by only counting the length or comparing the first digit. For scientific notation, we ensure 70% pairs of compared numbers with the same exponential part so that models cannot directly get the answer without comparing the significand part. For fractions, we ensure the numbers are both less than one, avoiding the model can just compare them with 1 to get more than 50% accuracy.

- For "to_float-Fraction", we require that the fraction can be converted into a finite decimal, that is the denominator contains only factors 2 and 5.

- For "add/sub-Scientific", we require the exponential part of each number to have a difference less than 5 to make sure that the generated answer will not be too long.

The pre-processing could introduce additional duplicated data, so we implement a post-filtering step to remove duplicates and ensure data integrity.

### A.1.8 METRICS

For digit match, we should first align the numbers. For the integers and integer parts in floats, the numerator and denominator of fractions, and the exponential part of the scientific notation, we use the right alignment. For the decimal part in floats (as well as the in the significand part in scientific notation), we use the left alignment.

For dlength, we first measure the difference of each part of a number and then add the absolute values up.

Besides the average metrics in each range, we also present the following metrics: *well-learned digits* and *performance-preserving digits* to demonstrate the model's upper and lower performance limits on length. These represent the maximum number of digits that can maintain over 90% and 10% accuracy, respectively. (For digit match, the thresholds are set to 90% and 50%, and for dlength, where smaller is better, the thresholds are 0.1 and 1).

We ensure that there is no duplicated sample in dataset, so for some range, the test samples could be less than 1000. We also omit 1 digit or some 2 digit test in our testbed to make sure that unit rules can be included in a training set.

### A.2 PROMPTS AND OTHER DETAILS TO TEST BASELINE MODELS

For all models in our test, we first provide a "format prompt" describing the expected return format (and avoiding models generating complex CoT), and a "task prompt" describing the task. We use some easy problems to ensure powerful models (gpt-4o-mini and Llama-3.1-8B) can correctly understand the tasks and expected return format by the prompts. The expected return representation of each task is referred to in Appendix A.1.4.

The **format prompt** based on the expected return type of the task is as follows:

- **Integer**: Directly return the answer as an integer without any comma separator, like 123 .
- **float**: Directly return the answer as a float without any comma separator, like 10.4 .
- **Fraction**: Directly return the answer as an **irreducible** fraction without any comma separator, like 7/13 .
- **Scientific Notation**: Directly return the answer as a scientific notation without any comma separator, like 1.23e4 . The float part should be in the range [1, 10).

The **task prompts** are listed as follows where <a> and  are numbers.

- **Add**: Add two numbers: <a> +  =
- **Sub**: Subtract two numbers: <a> -  =
- **Multiply**: Multiply two numbers: <a> *  =
- **Truediv**: Divide two numbers and return the result as a fraction. <a> /  =
- **Floordiv**: Divide two numbers and return the result as an integer. <a> //  =
- **Mod**: Divide two numbers and return the remainder. <a> % 
- **Max**: Get the maximal number: <a> and  =
- **Min**: Get the minimal number: <a> and  =
- **Digit max**: Compare two numbers digit by digit and return the larger digit at each position, treating any missing digits as 0. <a> and  =
- **Digit min**: Compare two numbers digit by digit and return the smaller digit at each position, treating any missing digits as 0. <a> and  =
- **Digit add**: The task is to add two given numbers digit by digit and return the result modulo 10 (ignoring carry), treating any missing digits as 0. <a> digit add  =
- **Get digit**: Get the digit at the given position (from left to right, starting from 0). <a> at position  =
- Length: The total number of digits of <a> =

- **Count**: Count the number of the given digit in the given number: `<a>` count the occurrence time of digit `` =
- **To_float**: Convert the number to float: `<a>` =
- **To_scient**: Convert the number to scientific notation: `<a>` =
- **Sig_fig**: Convert the number to scientific notation: `<a>` and keep significant figures as ``.

Notice that all prompts are ended with an "=" so that we can easily separate the input question and the generation of models. When we use the texts in supervised finetuning (SFT), the context before the "=" is not involved in the loss calculation.

For GPT-4o and GPT-4o-mini, we also add a system message as follows and use the aforementioned question as user message:

> You are a capable math assistant. Return your solution without any process in the format: The answer is [YOUR ANSWER]. The final answer must strictly match the format `<regex>`.

where the `<regex>` is a regular expression based on the expected return format:

- **Integer**: r"\d+"
- **Float**: r"\d+\.\d+"
- **Fraction**: r"\d+/\d+"
- **Scientific Notation**: r"\d+\.\d+e\d+"

We use the models expect GPT from huggingface and use the default tokenizer, model and generation configuration provided by the models. We test GPT-4o and GPT-4o-mini by the OpenAI API, where GPT-4o means gpt-4o-2024-0806 and GPT-4o-mini means GPT-4o-mini-2024-07-18. For Qwen2-72B and Llama-3.1-70B, we additionally use 4-bit quantization but we also test several samples without quantization and ensure this quantization does not affect generation quality.

We retrieve the first match of the corresponding regular expression after the "=" as the answer. If there is no retrieve, we use an empty answer to calculate the metrics, where exact match and digit match is both zero and the dlength is the total length of the groundtruth number.

## A.3    Full test results of LLMs

We show the full NUPA Test results in Figures 5 (exact match), 6 (digit match), 7 (dlength) and Table 7, 8, 9 (well-learned digits and performance-preserving digits for each metrics).

With the detailed metrics, we can more clearly understand the behavior of some models on some tasks. For example, we find that the "exact match" and "digit match" of some models like Qwen-2 and GPT-4o on the "integer-max" task are similar, suggesting that when the models know which one is correct, they can always copy the answer from question correctly. So the wrong answer comes from incorrect comparison. Another example is the Llama-2 performance on max-hard. Because the length of two input numbers and the groundtruth answer in the max-hard task are all the same, most models show less dlength on this task suggesting they know that "the answer should have the same length of inputs", but we find Llama-2 shows dlength approximately equal to the average length in the range, suggesting that Llama-2 cannot generate a valid answer on this task. These are just a few examples to illustrate how more detailed metrics can help us gain a deeper understanding of model behavior. There are many possible conclusions, but there are too many to list here.

### A.3.1    Few-shot learning

To ensure the output format of models is as precise as possible, we employ 5-shot learning. For each task, we select one sample from 5 different lengths respectively and test the few-shot performance. Table 10 summarizes the exact match score performance across three selected tasks and Figure 8 shows more tasks. Notably, providing an explicit output format results in general performance improvements across tasks and input lengths. In most tasks, the models can usually produce accurately formatted

outputs even in the zero-shot setting, with limited additional benefit observed from few-shot examples, while the few-shot examples can indeed provide some performance improvement.

We find that the conclusions mentioned in main paper have still holds. For example, the performance also significantly decreases as the length increases or the tasks and representations become unfamiliar (like Add-Fraction, Add-Scientific or floordiv). And the performance of digit-related tasks are still unsatisfying.

Table 10: Few-shot performance on selected tasks

| Model | Add Int | | | | Max Float | | | | Floordiv Int | | | |
|---|---|---|---|---|---|---|---|---|---|---|---|---|
| | S | M | L | XL | X | M | L | XL | X | M | L | XL |
| Llama-2-7b-hf-5-shot | 0.61 | 0.12 | 0.00 | 0.00 | 0.68 | 0.55 | 0.49 | 0.43 | 0.04 | 0.01 | 0.01 | 0.00 |
| Llama-2-7b-hf | 0.74 | 0.11 | 0.00 | 0.00 | 0.44 | 0.47 | 0.28 | 0.15 | 0.04 | 0.01 | 0.00 | 0.00 |
| Llama-3.1-8B-5-shot | 0.94 | 0.41 | 0.10 | 0.01 | 0.88 | 0.81 | 0.63 | 0.54 | 0.23 | 0.02 | 0.01 | 0.00 |
| Llama-3.1-8B | 0.95 | 0.38 | 0.06 | 0.02 | 0.70 | 0.57 | 0.41 | 0.36 | 0.19 | 0.01 | 0.01 | 0.01 |
| Qwen2-7B-5-shot | 0.83 | 0.82 | 0.37 | 0.04 | 1.00 | 0.98 | 0.81 | 0.64 | 0.28 | 0.08 | 0.03 | 0.01 |
| Qwen2-7B | 0.93 | 0.70 | 0.23 | 0.03 | 0.68 | 0.72 | 0.55 | 0.43 | 0.22 | 0.05 | 0.01 | 0.02 |

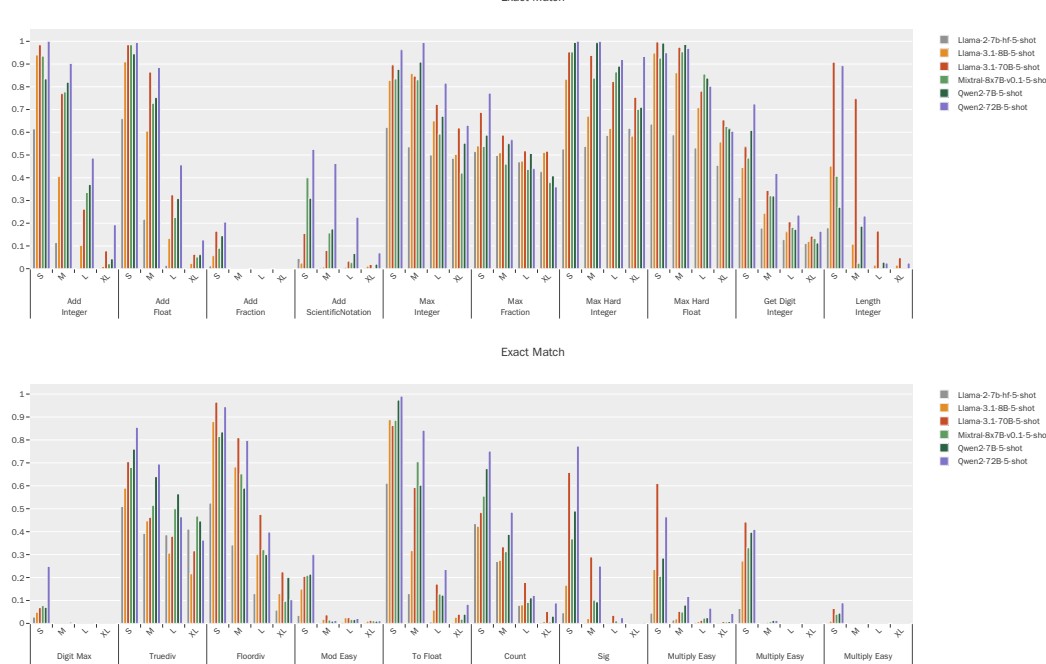

Figure 8: Parts of performance of state-of-the-art LLMs on NUPA benchmark with 5-shot examples.

## A.4 TOKENIZER, PE AND DATA FORMATS

### A.4.1 EXPERIMENT DETAILS

We train several models to test the effectiveness of tokenizers, PEs and data formats. Unless otherwise mentioned, our model architecture uses the Llama-3.1 architecture (Decoder-only Transformers with causal masking, autoregressive generation, and RoPE as the default PEs). We modify the layer numbers, hidden size and the number of heads to change the parameter size of models. See Table 11. We keep all hyperparameters, except model size, consistent with the original Llama setup in the implementation from Huggingface. We use the default sampling generation strategy with default hyperparameters, where the temperature is set as 0.6 and top_p is 0.9. About the meaning of these settings please refer to Llama technique report (Meta, 2024a) and model cards (Meta, 2024b).

To train these models, we use the AdamW optimizer (Loshchilov & Hutter, 2019) with a learning rate of 5e-5, weight decay of 0.01, and batch sizes of 256, 64, and 32 for 0.1B, 0.9B, and 3B models, respectively. Other optimizer settings follow the default values in the Transformers library. We sample 1e7 samples for *each* length (where feasible) and concatenate them into a single training

set. Models are trained for one epoch using a cosine decay learning rate scheduler, and the best checkpoint on validation data is reported.

Our experiments were conducted on a cluster equipped with Nvidia A800 GPUs (80GB memory). Training a 100M model takes 5–8 hours, a 1B model approximately 1 day, and a 3B model around 2 days on a single A800 GPU. Finetuning a pretrained model typically takes about 1 day.

Table 11: Detailed model settings for experiments.

| parameter size | num hidden layers | hidden size | intermediate size | num attention heads | num KV heads |
|---|---|---|---|---|---|
| 100M | 8 | 1024 | 3584 | 8 | 2 |
| 0.9B | 16 | 2048 | 7168 | 16 | 4 |
| 3.0B | 24 | 3072 | 10752 | 24 | 6 |

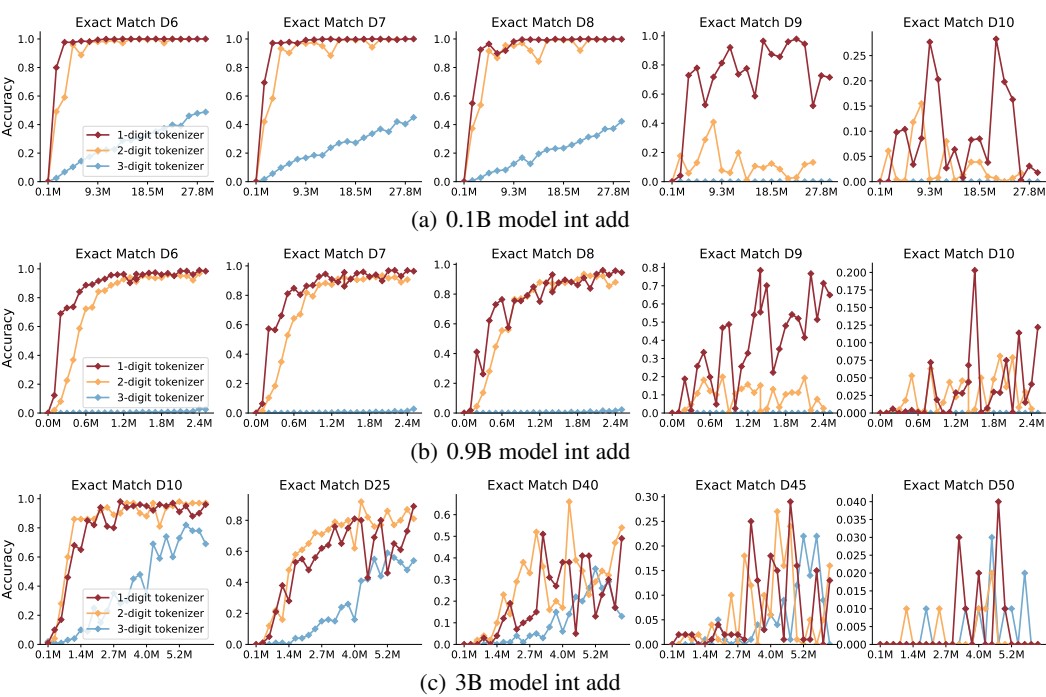

Figure 9: Accuracy of models of 0.1B, 0.9B and 3B parameters trained with 1-3 digit tokenizer on the task of integer addition. X-axis is the number of seen training samples.

### A.4.2 TOKENIZATION

We experiment on models of 3 different size, including 0.1B, 0.9B and 3B. For the 0.1B and 0.9B models, we train them on integer addition of 1-8 digits; for the 3B model, we train it on the same task of 1-40 digits.

Figure 9 illustrates the in-domain performance of these three models in the first three columns and their out-of-domain (OOD) performance in the last two columns. Here we use the exact match metric. In our experiments of the 0.1B and 0.9B models, the one-digit and the two-digit tokenizer demonstrate comparable performance in the in-domain test, while the one-digit tokenizer exceeds the others to a large extent in length generalization. In contrast, the three-digit tokenizer exhibits poor performance in both in-domain and out-of-domain evaluations. Tokenizers with an increasing number of digits significantly hinder subbillion models' NUPA. In the experiments of the 3B model, the two-digit tokenizer matches the one-digit tokenizer in both in-domain and OOD performance. In addition, the three-digit tokenizer shows the potential in length generalization for the first time, yet its performance remains inferior to that of the smaller tokenizers. This indicates that scaling up the model size

indeed alleviate the challenges in developing NUPA caused by larger tokenizers. Nevertheless, larger tokenizers do not present any distinct benefits in either in-domain or out-of-domain generalization in both small and large models.

We report the results according to different metrics from Figure 4 including digit match and dlength in Figure 10 and Figure 11.

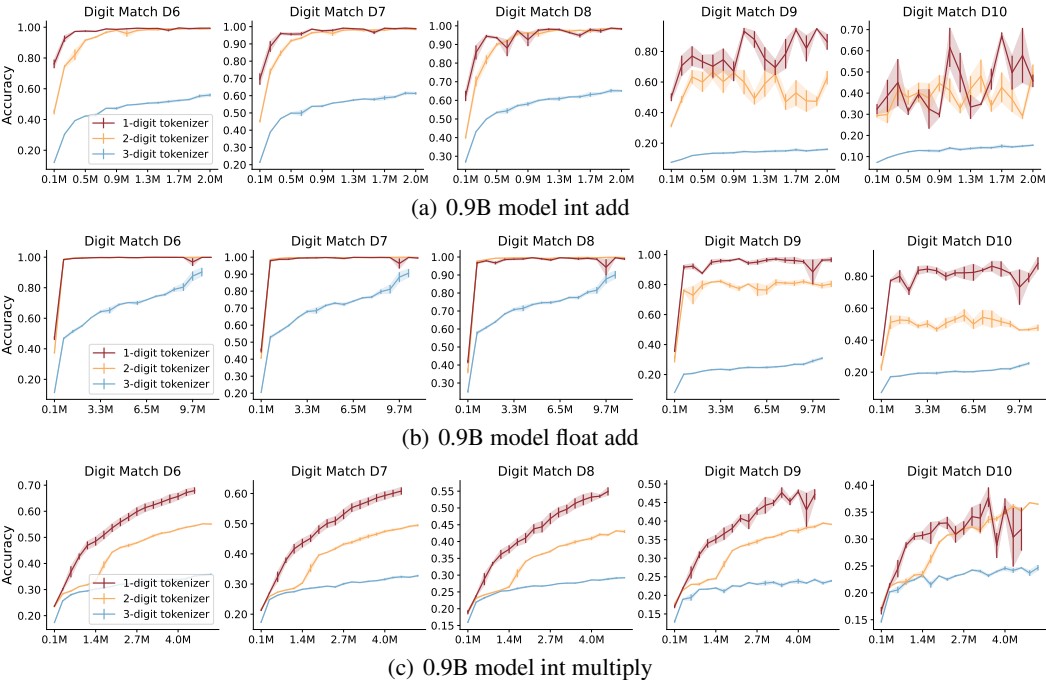

Figure 10: Accuracy of 0.9B models trained with 1-3 digit tokenizer on three tasks of integer addition, float addition and integer multiplication according to **digit match**. X-axis is the number of seen training samples.

**Random tokenizer** Introduced as "sub-word regularization" by Kudo (2018); Provilkov et al. (2020), the random tokenizer splits words like "Hello world" into variable tokens such as "He/llo/ world" or "Hell/o/ world". Though not widely used in LLMs, Sathe et al. (2024) found that it enhances reasoning by introducing variability in generation path. Inspired by this, we apply this to the numbers, segmenting numbers into tokens with lengths randomly chosen between 1 and a predefined maximum, instead of using greedy left-to-right segmentation.

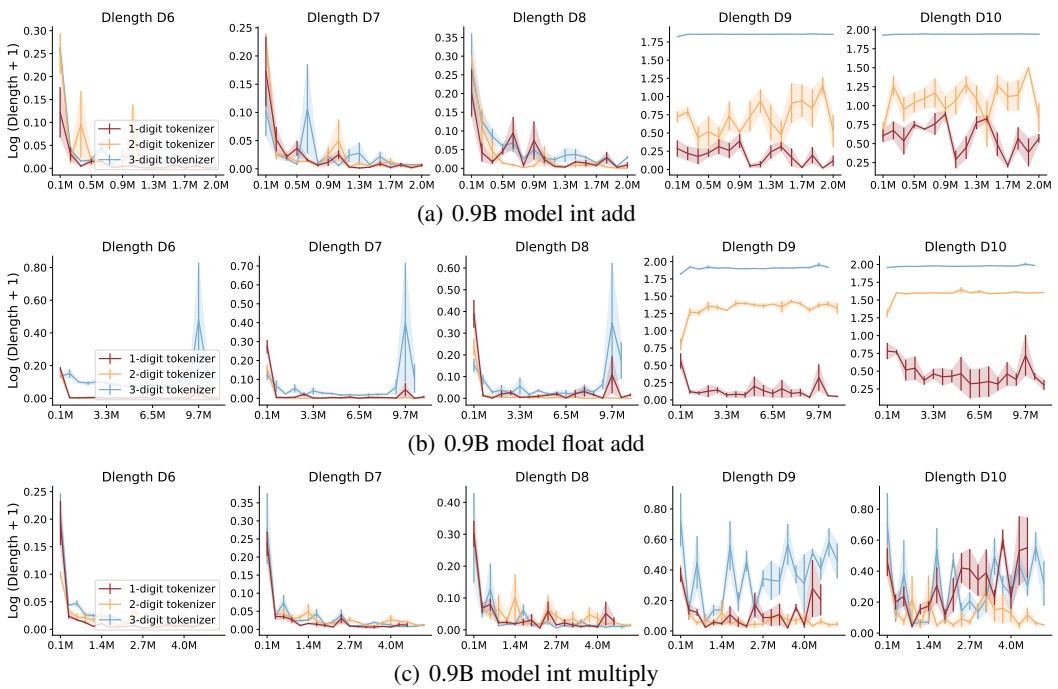

Figure 11: Accuracy of 0.9B models trained with 1-3 digit tokenizer on three tasks of integer addition, float addition and integer multiplication according to **dlength**. Here we report $log_2(\text{dlength} + 1)$.X-axis is the number of seen training samples.

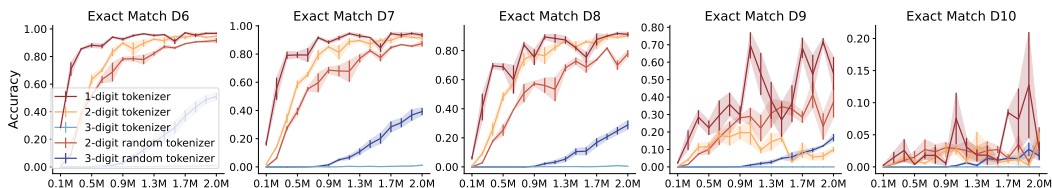

Figure 12: Accuracy of 0.9B models trained with 1- to 3- digit tokenizers and 2- to 3- digit random tokenizers on integer addition. Shadow shows the standard error. D$n$ means $n$ digits. X-axis is the number of seen training samples.

Figure 12 shows the performance of 1- to 3-digit tokenizers alongside 2- to 3-digit random tokenizers, where $n$-digit random tokenizer means the one with maximal length $n$. In terms of in-domain generalization, the three-digit random tokenizer outperforms the three-digit standard tokenizer, while the two-digit random tokenizer shows a slight decline compared to its standard counterpart. We believe this is because the 0.9B model is capable of learning the two-digit tokenizer well, and the added perturbation from random tokenization acts as a form of regularization, introducing noise that slightly affects performance. The random tokenizers consistently outperform their standard counterparts in OOD generalization, indicating the regularization benefits in that aspect. In the case of the three-digit tokenizer, which is more challenging for a 0.9B model to learn, random tokenization generates smaller tokens, making the learning process easier and leading to improved in-domain performance. However, they **still fall short of the performance achieved by the one-digit tokenizer**.

### A.4.3 PEs

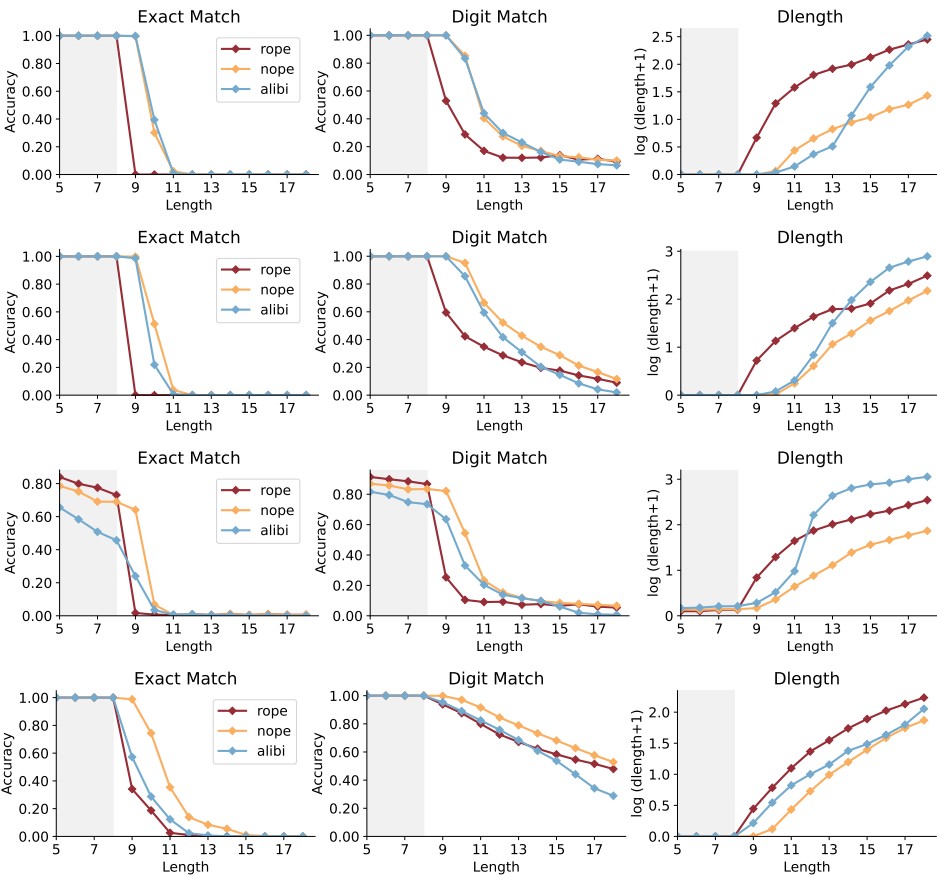

Figure 13: Exact match, digit match and dlength of 100M models trained with various PE, including RoPE, NoPE and Alibi. From top to bottom, the tasks are integer addition, float addition, fraction multiplication and scientific notation.

We show exact match, digit match and dlength of 100M models trained with various PE, including RoPE, NoPE and Alibi in Figure 13. We find NoPE and Alibi achieve better length generalization than RoPE, which is consistent with previous work like Zhou et al. (2024b).

To explain the mechanism of PEs, it is necessary to describe what the "generalization" is about. In most tasks, there is an intrinsic "*length-agnostic*" calculating rule, independent of the length of input numbers. For example, the addition rules: "*align numbers by their least significant digits, add them digit by digit and carry over if the sum exceeds 9*" is length-agnostic because it applies universally, regardless of the input length. However, during training on data with restricted length range (like 1 to 8), models may also learn *length-related* rules that fit the training data, such as combining normal addition rules with constraints like "*the output length must range from 1 to 8*". Because these two rules are *indistinguishable*, prior knowledge should be added into the model as an inductive bias to help the model learn the "length-agnostic" rules expected in most practical settings (Abbe et al., 2024; Chen et al., 2024).

Table 12: RoPE performance with *standard error* from three repeated experiments. D$n$ means $n$ digits where D8 is the longest in-domain length and D9 is the shortest out-of-domain length.

| | Exact Match | | Digit Match | | Dlength | |
|---|---|---|---|---|---|---|
| | D8 | D9 | D8 | D9 | D8 | D9 |
| Int-add | 1.00±0.00 | 0.00±0.00 | 1.00±0.00 | 0.45±0.02 | 0.00±0.00 | 1.07±0.02 |
| Float-add | 1.00±0.00 | 0.00±0.00 | 1.00±0.00 | 0.59±0.02 | 0.00±0.00 | 1.06±0.01 |
| Frac-mul | 0.70±0.01 | 0.01±0.00 | 0.85±0.01 | 0.22±0.02 | 0.18±0.02 | 1.45±0.08 |
| Sci-add | 1.00±0.00 | 0.23±0.08 | 1.00±0.00 | 0.92±0.01 | 0.00±0.00 | 0.66±0.11 |

According to our experiments, we find that (1) **RoPE encourages the model to rely on the length** of the input. The first evidence is that RoPE causes the model's predictive performance to plummet dramatically just beyond the training boundary. We report the RoPE's performance at the boundary of training length in Table 12 where D8 (digit 8) is the longest length in the training range while D9 (digit 9) is the shortest length out of the training range. In "*int-add*" task, the exact match drops from nearly 100% to 0% when moving from 8 to 9 digits, while "dlength" rises from 0 to 1.07 (Table 12). This indicates that the model has a significant probability of generating shorter results, avoiding the generation of more than 8-digit answers. At the same time, RoPE not only constrains the model's output length but also affects the digit pairing. The performance of 100% for inputs of 8 digits indicates that the model performs calculations for each position unless it can successfully align the corresponding digits. However, when the model encounters 9-digit inputs, digit match drops significantly to 50%, suggesting a considerable probability of failing to align the digits. Similar results on the other three tasks suggest that it is a **task-agnostic** behavior. The only exception is the *digit match* of *scientific notation addition*. We discuss the results later.

(2) On the other hand, **length learning provided by RoPE appears to be a shortcut**. In cases where the model is extremely small or has been trained very little, we see the advantages of this "*shortcut*". In Table 13, we train a 2-layer transformer (1.3M parameters) on integer addition using three different PEs on 1- to 8- digit integer addition or the 0.1B model with only 1M samples, we find RoPE shows the best in-domain performance. Experiments on

Table 13: 8-digit digit-match accuracy with small model or small dataset.

| | 1.3M Model | 1M Samples |
|---|---|---|
| RoPE | **0.091** | **0.97** |
| NoPE | 0.061 | 0.78 |
| Alibi | 0.056 | 0.23 |

the other three tasks are shown in Table 14 and Table 15, where the RoPE always surpasses others.

As a possible explanation about why Alibi and NoPE achieve better length generalization, our experiments suggest that for length generalization in number tasks, the required inductive bias is to interpret the input as a sequence of digits while deliberately ignoring its overall length. RoPE, as a positional encoding that enables the model to quickly learn position-related information, may lead the model to adopt a length-dependent shortcut (Table 13), causing it to favor length-related rules. In contrast, both Alibi and NoPE diminish this reliance on position and length, encouraging the model to treat each unit's operation as a step-by-step process, thereby achieving better length generalization.

**Discuss about scientific addition**   The results in Table 12 reveal a clear trend where performance drops from 8-digit to 9-digit numbers, with one exception: the digit match score in the scientific notation addition task, which remains relatively high at 0.93 even for 9-digit numbers. We believe it is mainly because of the alignment mechanism between two scientific notations which differs from other representations. In other representations, numbers are aligned by **position** — integers from the most-left digit and the floats by the decimal point. However, in scientific notation, alignment depends on the difference in exponent values, which reduces RoPE's reliance on position and mitigates length overfitting. Despite this, the effect of RoPE limiting output length remains apparent, as evidenced by the significant increase in the dlength score.

### A.4.4 DATA FORMATS

Table 16: Exact match of 0.1B models trained on integer addition and float addition respectively with various compositions of reverse formatting and zero padding.

| | Integer Addition | | | | Float Addition | | | | | | | | | |
|---|---|---|---|---|---|---|---|---|---|---|---|---|---|---|
| | rev | rev +pad | no | pad | rev total | rev total + pad | rev each | rev each + pad | rev dec | rev dec + pad | rev int | rev int + pad | no | pad |
| d9 | 0.97±.05 | **1.00**±.00 | 0.98±.02 | **1.00**±.01 | 0.11±.01 | 0.24±.00 | 0.12±.00 | 0.24±.00 | 0.12±.00 | 0.24±.00 | **1.00**±.00 | **1.00**±.00 | 0.99±.01 | **1.00**±.00 |
| d10 | 0.69±.11 | **0.91**±.05 | 0.16±.11 | 0.50±.34 | 0.07±.03 | 0.21±.01 | 0.10±.02 | 0.23±.00 | 0.07±.02 | 0.17±.04 | **0.97**±.03 | 0.87±.16 | 0.17±.04 | 0.76±.19 |

We provide the experiments in Table 16 and the evaluation curves of compositions of reverse formatting, zero padding and index hints in Figure 14, Figure 15, Figure 16, Figure 17 and Figure 18. We experiment on 0.1B models trained on 1- to 8- digit training samples. Here we all use the exact match metric.

Previous work (Zhou et al., 2024b) believes that reverse formatting can help the calculation of each digit by aligning the calculation order to the right-to-left order that humans are accustomed to and solve the carrying problem. That is, from left-to-right, we cannot determine the result of the current digit unless the next digit results and whether there is a carrying have been known. However, a more detailed analysis can explain why the order is not as important as previously believed:

Regarding addition, the cases where reverse formatting can make a difference through the effects of assisting carry-over calculations are quite rare. Most of the time, knowing the result of the next digit allows us to determine the answer for the current digit. When the next digit addition is not less than 10 (without considering further carrying from the following digit), there must be a carrying from that digit into the current one, no matter what the result of the later digits is. And when the next digit addition is not more than 8, there will never be a carrying. The only exception is the next digit addition is 9. In this situation, we must refer to the next two digits to determine the current digit results. Therefore, we point out that, although in the worst-case scenario, performing non-reversed addition requires $O(n)$-length looking forward for each digit ($44445 + 55556 = 100001$), and reversing could solve this problem, such cases are extremely rare. In most instances, the task can be accomplished with a very limited *local view*.

About the experiments of index hint, we show in Table 17. Our conclusion on index hints seems to contradict the findings of Zhou et al. (2024b), where models with index hints appeared to achieve better results. We believe this discrepancy may be related to model size and digit range. In their work, a much smaller model (only 25M parameters) was used, but the training range covered 1-40 digits. This reduced the model's ability to learn the patterns independently without external hints, resulting in a different learning outcome where the model began to rely on index hints. As a piece of evidence, when Zhou et al. (2024b) train 1-10 digits, the performance without index hint is OK. (But they did not provide the complete results of 1-10 digit training in their work.) The effectiveness of index hints may involve complex interactions, which could be an interesting direction for future research.

### A.4.5 NUPA FINETUNING WITH PE, TOKENIZER AND REPRESENTATION MODIFICATION

We show parts of results of our attempt to finetune a Llama-3.1-8B model with PE, tokenizer and data format modification in Table 18. All the checkpoint we select by the lowest valid loss. No one can outperform the naive finetuning or the original Llama.

### A.5 RULE-FOLLOWING CHAIN-OF-THOUGHT

### A.5.1 DATA FORMAT OF RULE-FOLLOWING CoT

Rule-following CoT consists of a rule prompt and a response to follow the rule. To generate the CoT samples for a task, first the computational rules are converted to Python code. We utilize a code LLM to write the code in this experiment. Then, a data generation script which can generate the rule-following intermediate steps for any given input is manually written. With the script, given a pair of numbers and a task, both the rules (code prompt) and the response can be generated automatically. In the finetuning stage, both the prompt and the response are provided to the model but only the loss on the response part is calculated. During inference, the code prompt is still provided and the models are expected to generate the rule-following steps as well as the final answer.

Below is an example of a complete Rule-Following CoT data format, where the model is required to solve integer addition task with a right-to-left recursion and three digit addition as the unit task.

Prompt:

Follow the given rule to solve the question.
Rule:

```python
def add(num1, num2):
    result = ''
    carry = 0
    # Main Loop
    while num1 or num2:
        digit1 = int(num1[-3:]) if num1 else 0
        digit2 = int(num2[-3:]) if num2 else 0
        total = digit1 + digit2 + carry
        result = str(total%1000) + result
        carry = total//1000
        num1 = num1[:-3] if num1 else num1
        num2 = num2[:-3] if num2 else num2
    if carry:
        result = str(carry) + result
    result = result.lstrip('0') or '0'
    return result
```

Q: Return the answer as an integer without any delimiter, like 123. Add two numbers: 28 + 35

Response:

```
num1 = "28"
num2 = "35"
result = ""
carry = 0

while num1 or num2:

check the stop criterion
enter the loop

digit1 = int(num1[-3:]) if num1 else 0
digit2 = int(num2[-3:]) if num2 else 0

num1 = "28"
num2 = "35"
digit1 = 28
digit2 = 35

total = digit1 + digit2 + carry

carry = 0
total = 28 + 35 + 0 = 63

result = str(total%1000) + result
carry = total//1000
```

```
total % 1000 = 63%1000 = 63
result = "063" + "" = "063"
carry = 0//1000 = 0

num1 = num1[:-3]
num2 = num2[:-3]

num1 = ""
num2 = ""

while num1 or num2

check the stop criterion
num1 = "" and num2 = ""
end the loop

if carry:
    result = str(carry) + result

carry = 0
pass

result = result.lstrip("0") or "0"

result = "63"

return result

return "63"
So the answer is 63.
```

### A.5.2 MAXIMUM DIGIT LENGTHS WITHIN CONTEXT WINDOW

The selective tasks used to train the RFFT are shown in Table 19 and we also report the maximal length within 2k tokens context windows limitation.

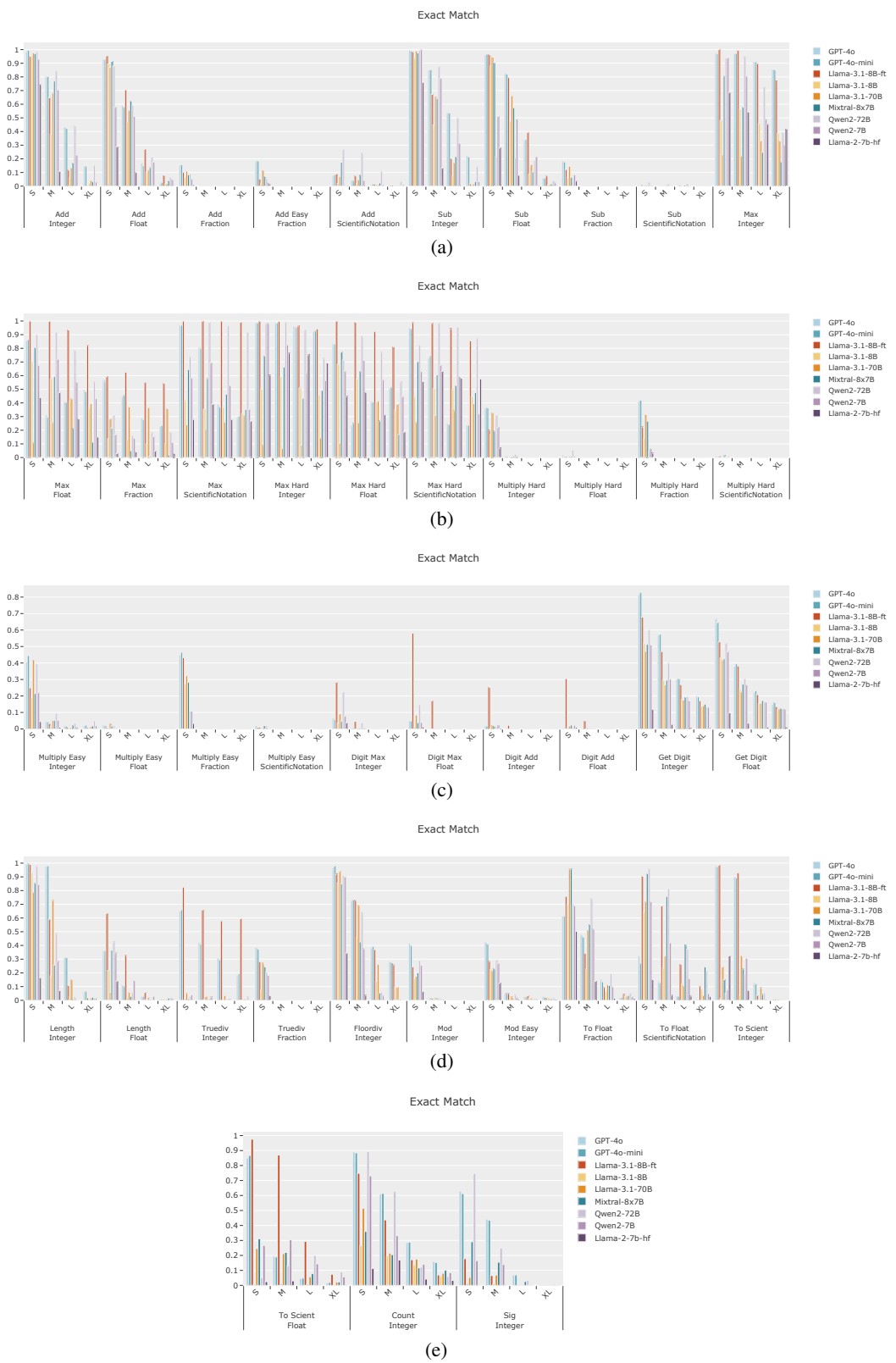

Figure 5: Exact match of models tested on NUPA Test.

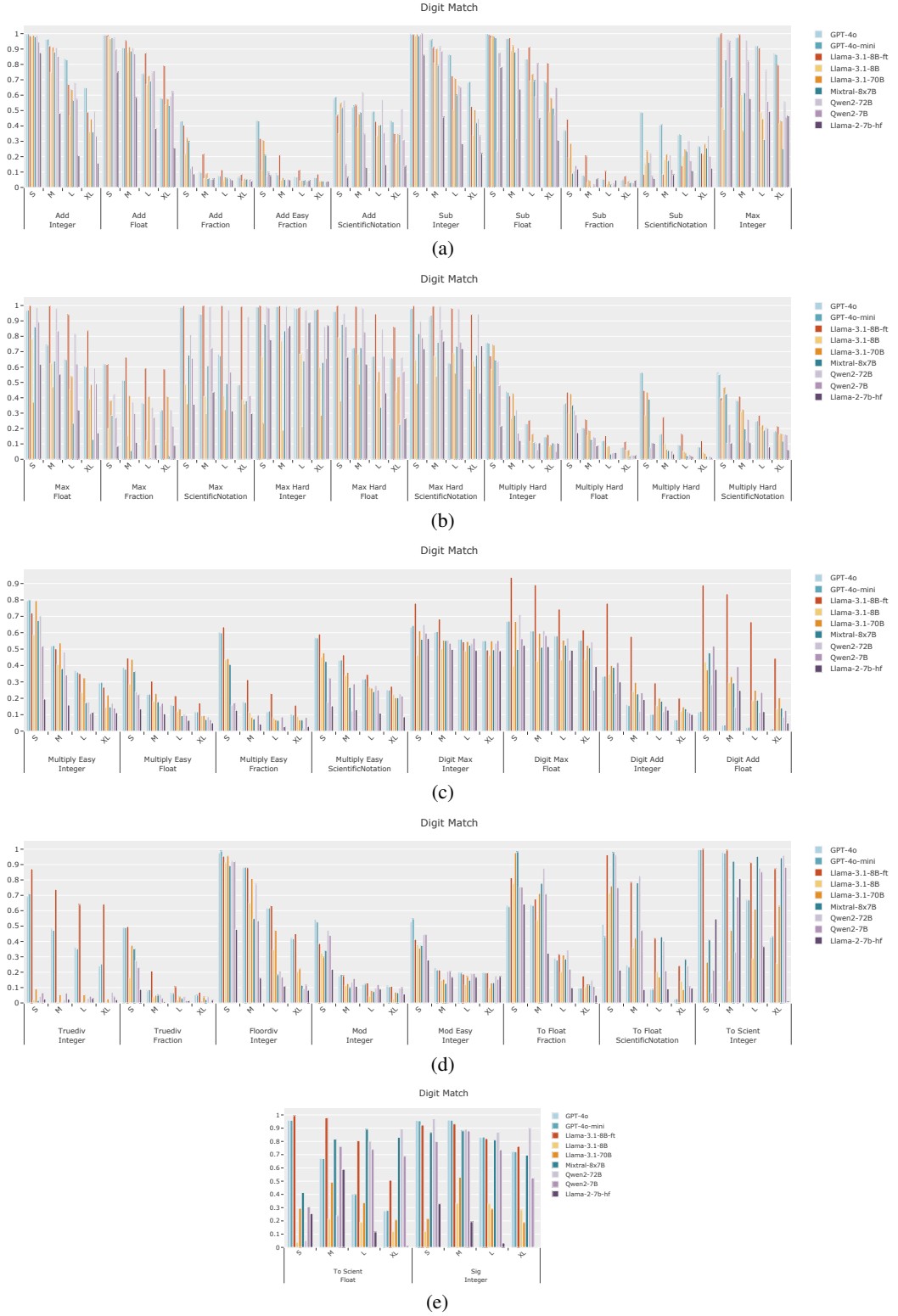

Figure 6: Digit match of models tested on NUPA Test.

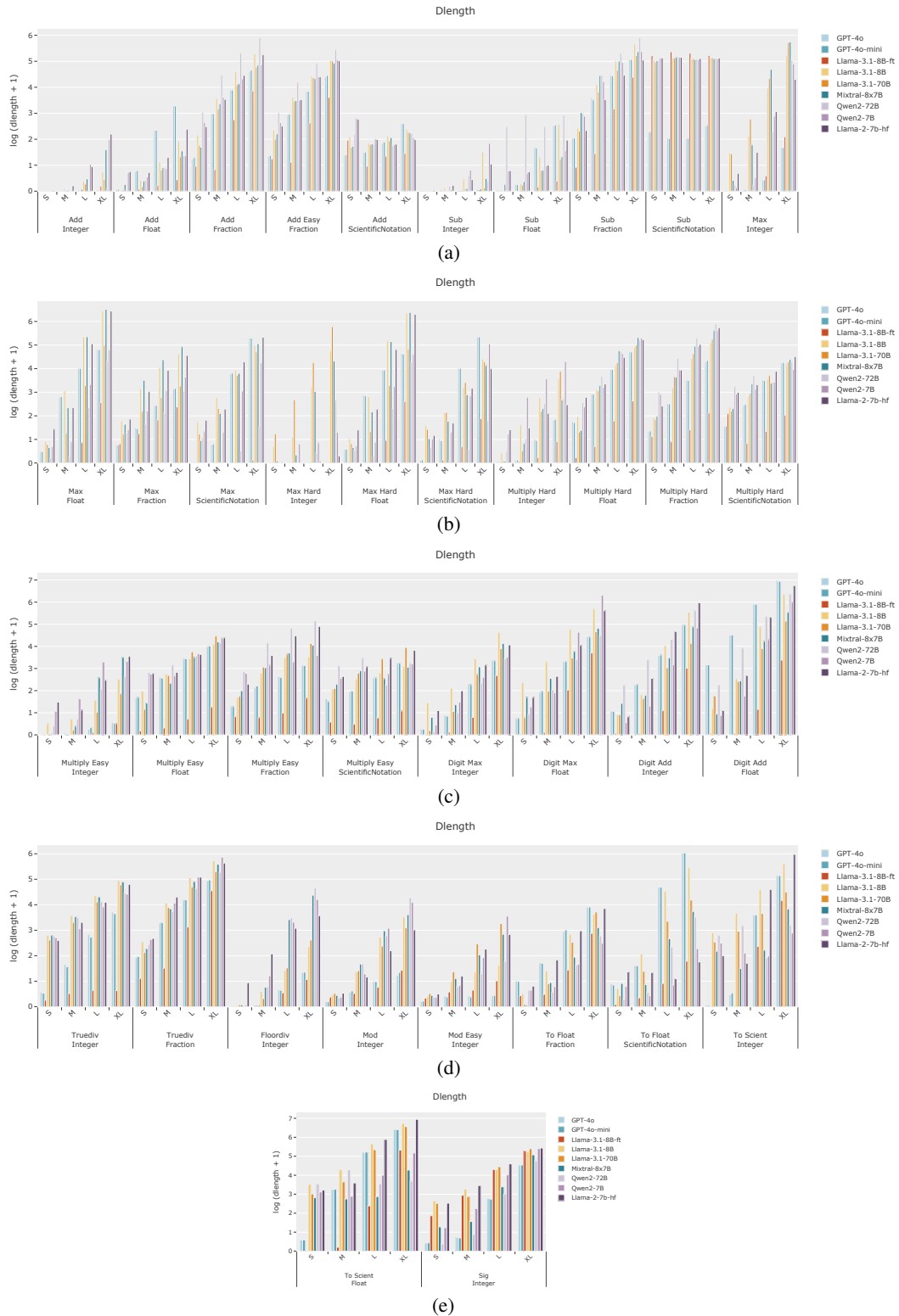

Figure 7: Dlength of models tested on NUPA Test. Note that we use $\log_2(\mathbf{dlength} + 1)$ as the ylabel in the figure.

Table 7: Well-learned digits / performance-preserving digits of models tested on NUPA Test according to exact match.

| | Add Int | Add Float | Add Frac | Add Easy Frac | Add Sci | Sub Int | Sub Float | Sub Frac |
|---|---|---|---|---|---|---|---|---|
| GPT-4o-mini | 5 / 20 | 4 / 11 | 0 / 1 | 0 / 2 | 0 / 0 | 6 / 20 | 5 / 15 | 0 / 1 |
| GPT-4o | 5 / 20 | 4 / 11 | 0 / 1 | 0 / 1 | 0 / 0 | 6 / 20 | 4 / 15 | 0 / 1 |
| Qwen2-72B | 6 / 20 | 0 / 15 | 0 / 1 | 0 / 1 | 0 / 11 | 6 / 20 | 0 / 15 | 0 / 1 |
| Qwen2-7B | 4 / 14 | 0 / 15 | 0 / 1 | 0 / 0 | 0 / 7 | 4 / 14 | 0 / 13 | 0 / 1 |
| Llama-3.1-8B-ft | 4 / 12 | 5 / 17 | 0 / 1 | 0 / 1 | 0 / 4 | 4 / 12 | 4 / 17 | 0 / 2 |
| Llama-3.1-70B | 4 / 15 | 0 / 11 | 0 / 1 | 0 / 1 | 0 / 0 | 6 / 11 | 4 / 11 | 0 / 1 |
| Llama-3.1-8B | 4 / 9 | 3 / 11 | 0 / 1 | 0 / 0 | 0 / 0 | 3 / 10 | 3 / 10 | 0 / 1 |
| Mixtral-8x7B | 5 / 10 | 4 / 11 | 0 / 1 | 0 / 1 | 0 / 6 | 4 / 15 | 3 / 11 | 0 / 1 |
| Llama-2-7b-hf | 0 / 6 | 0 / 7 | 0 / 0 | 0 / 0 | 0 / 0 | 0 / 6 | 0 / 5 | 0 / 1 |

| | Sub Sci | Max Int | Max Float | Max Frac | Max Sci | Max Hard Int | Max Hard Float | Max Hard Sci |
|---|---|---|---|---|---|---|---|---|
| GPT-4o-mini | 5 / 20 | 4 / 11 | 0 / 1 | 0 / 2 | 0 / 0 | 6 / 20 | 5 / 15 | 0 / 1 |
| GPT-4o | 5 / 20 | 4 / 11 | 0 / 1 | 0 / 1 | 0 / 0 | 6 / 20 | 4 / 15 | 0 / 1 |
| Qwen2-72B | 6 / 20 | 0 / 15 | 0 / 1 | 0 / 1 | 0 / 11 | 6 / 20 | 0 / 15 | 0 / 1 |
| Qwen2-7B | 4 / 14 | 0 / 15 | 0 / 1 | 0 / 0 | 0 / 7 | 4 / 14 | 0 / 13 | 0 / 1 |
| Llama-3.1-8B-ft | 4 / 12 | 5 / 17 | 0 / 1 | 0 / 1 | 0 / 4 | 4 / 12 | 4 / 17 | 0 / 2 |
| Llama-3.1-70B | 4 / 15 | 0 / 11 | 0 / 1 | 0 / 1 | 0 / 0 | 6 / 11 | 4 / 11 | 0 / 1 |
| Llama-3.1-8B | 4 / 9 | 3 / 11 | 0 / 1 | 0 / 0 | 0 / 0 | 3 / 10 | 3 / 10 | 0 / 1 |
| Mixtral-8x7B | 5 / 10 | 4 / 11 | 0 / 1 | 0 / 1 | 0 / 6 | 4 / 15 | 3 / 11 | 0 / 1 |
| Llama-2-7b-hf | 0 / 6 | 0 / 7 | 0 / 0 | 0 / 0 | 0 / 0 | 0 / 6 | 0 / 5 | 0 / 1 |

| | Multiply Hard Int | Multiply Hard Float | Multiply Hard Frac | Multiply Hard Sci | Multiply Easy Int | Multiply Easy Float | Multiply Easy Frac | Multiply Easy Sci |
|---|---|---|---|---|---|---|---|---|
| GPT-4o-mini | 5 / 20 | 4 / 11 | 0 / 1 | 0 / 2 | 0 / 0 | 6 / 20 | 5 / 15 | 0 / 1 |
| GPT-4o | 5 / 20 | 4 / 11 | 0 / 1 | 0 / 1 | 0 / 0 | 6 / 20 | 4 / 15 | 0 / 1 |
| Qwen2-72B | 6 / 20 | 0 / 15 | 0 / 1 | 0 / 1 | 0 / 11 | 6 / 20 | 0 / 15 | 0 / 1 |
| Qwen2-7B | 4 / 14 | 0 / 15 | 0 / 1 | 0 / 0 | 0 / 7 | 4 / 14 | 0 / 13 | 0 / 1 |
| Llama-3.1-8B-ft | 4 / 12 | 5 / 17 | 0 / 1 | 0 / 1 | 0 / 4 | 4 / 12 | 4 / 17 | 0 / 2 |
| Llama-3.1-70B | 4 / 15 | 0 / 11 | 0 / 1 | 0 / 1 | 0 / 0 | 6 / 11 | 4 / 11 | 0 / 1 |
| Llama-3.1-8B | 4 / 9 | 3 / 11 | 0 / 1 | 0 / 0 | 0 / 0 | 3 / 10 | 3 / 10 | 0 / 1 |
| Mixtral-8x7B | 5 / 10 | 4 / 11 | 0 / 1 | 0 / 1 | 0 / 6 | 4 / 15 | 3 / 11 | 0 / 1 |
| Llama-2-7b-hf | 0 / 6 | 0 / 7 | 0 / 0 | 0 / 0 | 0 / 0 | 0 / 6 | 0 / 5 | 0 / 1 |

| | Digit Max Int | Digit Max Float | Digit Add Int | Digit Add Float | Get Digit Int | Get Digit Float | Length Int | Length Float |
|---|---|---|---|---|---|---|---|---|
| GPT-4o-mini | 5 / 20 | 4 / 11 | 0 / 1 | 0 / 2 | 0 / 0 | 6 / 20 | 5 / 15 | 0 / 1 |
| GPT-4o | 5 / 20 | 4 / 11 | 0 / 1 | 0 / 1 | 0 / 0 | 6 / 20 | 4 / 15 | 0 / 1 |
| Qwen2-72B | 6 / 20 | 0 / 15 | 0 / 1 | 0 / 1 | 0 / 11 | 6 / 20 | 0 / 15 | 0 / 1 |
| Qwen2-7B | 4 / 14 | 0 / 15 | 0 / 1 | 0 / 0 | 0 / 7 | 4 / 14 | 0 / 13 | 0 / 1 |
| Llama-3.1-8B-ft | 4 / 12 | 5 / 17 | 0 / 1 | 0 / 1 | 0 / 4 | 4 / 12 | 4 / 17 | 0 / 2 |
| Llama-3.1-70B | 4 / 15 | 0 / 11 | 0 / 1 | 0 / 1 | 0 / 0 | 6 / 11 | 4 / 11 | 0 / 1 |
| Llama-3.1-8B | 4 / 9 | 3 / 11 | 0 / 1 | 0 / 0 | 0 / 0 | 3 / 10 | 3 / 10 | 0 / 1 |
| Mixtral-8x7B | 5 / 10 | 4 / 11 | 0 / 1 | 0 / 1 | 0 / 6 | 4 / 15 | 3 / 11 | 0 / 1 |
| Llama-2-7b-hf | 0 / 6 | 0 / 7 | 0 / 0 | 0 / 0 | 0 / 0 | 0 / 6 | 0 / 5 | 0 / 1 |

| | Truediv Int | Truediv Frac | Floordiv Int | Mod Int | Mod Easy Int | To Float Frac | To Float Sci | To Scient Int |
|---|---|---|---|---|---|---|---|---|
| GPT-4o-mini | 5 / 20 | 4 / 11 | 0 / 1 | 0 / 2 | 0 / 0 | 6 / 20 | 5 / 15 | 0 / 1 |
| GPT-4o | 5 / 20 | 4 / 11 | 0 / 1 | 0 / 1 | 0 / 0 | 6 / 20 | 4 / 15 | 0 / 1 |
| Qwen2-72B | 6 / 20 | 0 / 15 | 0 / 1 | 0 / 1 | 0 / 11 | 6 / 20 | 0 / 15 | 0 / 1 |
| Qwen2-7B | 4 / 14 | 0 / 15 | 0 / 1 | 0 / 0 | 0 / 7 | 4 / 14 | 0 / 13 | 0 / 1 |
| Llama-3.1-8B-ft | 4 / 12 | 5 / 17 | 0 / 1 | 0 / 1 | 0 / 4 | 4 / 12 | 4 / 17 | 0 / 2 |
| Llama-3.1-70B | 4 / 15 | 0 / 11 | 0 / 1 | 0 / 1 | 0 / 0 | 6 / 11 | 4 / 11 | 0 / 1 |
| Llama-3.1-8B | 4 / 9 | 3 / 11 | 0 / 1 | 0 / 0 | 0 / 0 | 3 / 10 | 3 / 10 | 0 / 1 |
| Mixtral-8x7B | 5 / 10 | 4 / 11 | 0 / 1 | 0 / 1 | 0 / 6 | 4 / 15 | 3 / 11 | 0 / 1 |
| Llama-2-7b-hf | 0 / 6 | 0 / 7 | 0 / 0 | 0 / 0 | 0 / 0 | 0 / 6 | 0 / 5 | 0 / 1 |

| | To Scient Float | Count Int | Sig Int |
|---|---|---|---|
| GPT-4o-mini | 5 / 20 | 4 / 11 | 0 / 1 |
| GPT-4o | 5 / 20 | 4 / 11 | 0 / 1 |
| Qwen2-72B | 6 / 20 | 0 / 15 | 0 / 1 |
| Qwen2-7B | 4 / 14 | 0 / 15 | 0 / 1 |
| Llama-3.1-8B-ft | 4 / 12 | 5 / 17 | 0 / 1 |
| Llama-3.1-70B | 4 / 15 | 0 / 11 | 0 / 1 |
| Llama-3.1-8B | 4 / 9 | 3 / 11 | 0 / 1 |
| Mixtral-8x7B | 5 / 10 | 4 / 11 | 0 / 1 |
| Llama-2-7b-hf | 0 / 6 | 0 / 7 | 0 / 0 |

Table 8: Well-learned digits / performance-preserving digits of models tested on NUPA Test according to digit match.

| | Add Int | Add Float | Add Frac | Add Easy Frac | Add Sci | Sub Int | Sub Float | Sub Frac |
|---|---|---|---|---|---|---|---|---|
| GPT-4o-mini | 9 / 20 | 6 / 20 | 0 / 2 | 0 / 2 | 0 / 14 | 11 / 20 | 9 / 20 | 0 / 1 |
| GPT-4o | 10 / 20 | 6 / 20 | 0 / 1 | 0 / 2 | 0 / 14 | 11 / 20 | 10 / 20 | 0 / 1 |
| Qwen2-72B | 7 / 16 | 7 / 20 | 0 / 0 | 0 / 0 | 0 / 20 | 6 / 15 | 0 / 16 | 0 / 0 |
| Qwen2-7B | 4 / 12 | 3 / 20 | 0 / 0 | 0 / 0 | 0 / 0 | 6 / 14 | 6 / 20 | 0 / 0 |
| Llama-3.1-8B-ft | 6 / 16 | 9 / 20 | 0 / 1 | 0 / 0 | 0 / 7 | 6 / 19 | 12 / 20 | 0 / 1 |
| Llama-3.1-70B | 6 / 15 | 7 / 20 | 0 / 1 | 0 / 1 | 0 / 4 | 6 / 17 | 7 / 20 | 0 / 1 |
| Llama-3.1-8B | 4 / 9 | 6 / 18 | 0 / 0 | 0 / 0 | 0 / 0 | 6 / 11 | 6 / 17 | 0 / 0 |
| Mixtral-8x7B | 6 / 12 | 5 / 18 | 0 / 1 | 0 / 1 | 0 / 6 | 5 / 15 | 5 / 18 | 0 / 0 |
| Llama-2-7b-hf | 3 / 6 | 0 / 8 | 0 / 0 | 0 / 0 | 0 / 0 | 3 / 5 | 0 / 9 | 0 / 0 |

| | Sub Sci | Max Int | Max Float | Max Frac | Max Sci | Max Hard Int | Max Hard Float | Max Hard Sci |
|---|---|---|---|---|---|---|---|---|
| GPT-4o-mini | 0 / 3 | 100 / 100 | 10 / 100 | 0 / 7 | 19 / 98 | 100 / 100 | 8 / 100 | 18 / 100 |
| GPT-4o | 0 / 0 | 100 / 100 | 10 / 100 | 0 / 7 | 19 / 98 | 100 / 100 | 8 / 100 | 16 / 100 |
| Qwen2-72B | 0 / 0 | 21 / 100 | 30 / 100 | 0 / 4 | 100 / 100 | 82 / 100 | 32 / 100 | 100 / 100 |
| Qwen2-7B | 0 / 0 | 11 / 86 | 10 / 98 | 0 / 0 | 0 / 82 | 13 / 100 | 7 / 100 | 6 / 69 |
| Llama-3.1-8B-ft | 0 / 0 | 83 / 100 | 75 / 100 | 0 / 20 | 100 / 100 | 100 / 100 | 79 / 100 | 100 / 100 |
| Llama-3.1-70B | 0 / 0 | 0 / 86 | 0 / 98 | 0 / 4 | 0 / 0 | 0 / 0 | 0 / 100 | 0 / 100 |
| Llama-3.1-8B | 0 / 0 | 0 / 67 | 0 / 93 | 0 / 0 | 0 / 54 | 0 / 100 | 0 / 96 | 0 / 100 |
| Mixtral-8x7B | 0 / 0 | 7 / 21 | 3 / 19 | 0 / 0 | 0 / 62 | 20 / 100 | 4 / 25 | 0 / 100 |
| Llama-2-7b-hf | 0 / 0 | 0 / 100 | 0 / 22 | 0 / 0 | 0 / 17 | 99 / 100 | 0 / 38 | 0 / 100 |

| | Multiply Hard Int | Multiply Hard Float | Multiply Hard Frac | Multiply Hard Sci | Multiply Easy Int | Multiply Easy Float | Multiply Easy Frac | Multiply Easy Sci |
|---|---|---|---|---|---|---|---|---|
| GPT-4o-mini | 0 / 5 | 0 / 0 | 1 / 2 | 0 / 4 | 0 / 6 | 0 / 0 | 1 / 2 | 0 / 4 |
| GPT-4o | 0 / 5 | 0 / 0 | 1 / 2 | 0 / 4 | 0 / 6 | 0 / 0 | 1 / 3 | 0 / 4 |
| Qwen2-72B | 0 / 5 | 0 / 0 | 0 / 0 | 0 / 0 | 0 / 6 | 0 / 0 | 0 / 0 | 0 / 0 |
| Qwen2-7B | 0 / 3 | 0 / 0 | 0 / 0 | 0 / 0 | 0 / 4 | 0 / 0 | 0 / 0 | 0 / 0 |
| Llama-3.1-8B-ft | 0 / 4 | 0 / 0 | 0 / 3 | 0 / 4 | 0 / 6 | 0 / 3 | 1 / 3 | 0 / 5 |
| Llama-3.1-70B | 0 / 5 | 0 / 3 | 0 / 2 | 0 / 3 | 0 / 6 | 0 / 3 | 0 / 2 | 0 / 3 |
| Llama-3.1-8B | 0 / 4 | 0 / 0 | 0 / 0 | 0 / 0 | 0 / 5 | 0 / 0 | 0 / 1 | 0 / 0 |
| Mixtral-8x7B | 0 / 4 | 0 / 0 | 0 / 1 | 0 / 0 | 0 / 5 | 0 / 0 | 0 / 2 | 0 / 0 |
| Llama-2-7b-hf | 0 / 0 | 0 / 0 | 0 / 0 | 0 / 0 | 0 / 0 | 0 / 0 | 0 / 0 | 0 / 0 |

| | Digit Max Int | Digit Max Float | Digit Add Int | Digit Add Float | Truediv Int | Truediv Frac | Floordiv Int | Mod Int |
|---|---|---|---|---|---|---|---|---|
| GPT-4o-mini | 0 / 100 | 0 / 100 | 0 / 0 | 0 / 0 | 0 / 5 | 1 / 2 | 5 / 15 | 0 / 3 |
| GPT-4o | 0 / 100 | 0 / 100 | 0 / 0 | 0 / 0 | 0 / 6 | 1 / 2 | 5 / 15 | 0 / 3 |
| Qwen2-72B | 0 / 100 | 0 / 100 | 0 / 0 | 0 / 0 | 0 / 0 | 0 / 1 | 3 / 8 | 0 / 3 |
| Qwen2-7B | 0 / 100 | 0 / 33 | 0 / 5 | 0 / 8 | 0 / 0 | 0 / 1 | 3 / 6 | 0 / 3 |
| Llama-3.1-8B-ft | 0 / 100 | 13 / 100 | 0 / 20 | 5 / 73 | 3 / 20 | 0 / 1 | 6 / 18 | 0 / 3 |
| Llama-3.1-70B | 0 / 100 | 0 / 100 | 0 / 0 | 0 / 0 | 0 / 0 | 0 / 1 | 4 / 12 | 0 / 0 |
| Llama-3.1-8B | 0 / 92 | 0 / 0 | 0 / 0 | 0 / 0 | 0 / 0 | 0 / 0 | 3 / 7 | 0 / 0 |
| Mixtral-8x7B | 0 / 100 | 0 / 97 | 0 / 5 | 0 / 5 | 0 / 0 | 0 / 1 | 3 / 7 | 0 / 3 |
| Llama-2-7b-hf | 0 / 100 | 0 / 42 | 0 / 0 | 0 / 0 | 0 / 0 | 0 / 0 | 0 / 3 | 0 / 0 |

| | Mod Easy Int | To Float Frac | To Float Sci | To Scient Int | To Scient Float | Sig Int |
|---|---|---|---|---|---|---|
| GPT-4o-mini | 0 / 3 | 0 / 8 | 0 / 9 | 23 / 67 | 8 / 24 | 31 / 100 |
| GPT-4o | 0 / 3 | 0 / 8 | 0 / 9 | 23 / 71 | 8 / 28 | 31 / 100 |
| Qwen2-72B | 0 / 3 | 6 / 9 | 16 / 28 | 100 / 100 | 96 / 100 | 100 / 100 |
| Qwen2-7B | 0 / 3 | 4 / 8 | 6 / 22 | 100 / 100 | 0 / 100 | 19 / 100 |
| Llama-3.1-8B-ft | 0 / 3 | 3 / 8 | 10 / 36 | 95 / 100 | 25 / 83 | 19 / 100 |
| Llama-3.1-70B | 0 / 0 | 4 / 9 | 0 / 12 | 0 / 100 | 0 / 21 | 0 / 21 |
| Llama-3.1-8B | 0 / 0 | 3 / 6 | 0 / 12 | 0 / 0 | 0 / 0 | 0 / 0 |
| Mixtral-8x7B | 0 / 3 | 4 / 9 | 14 / 36 | 100 / 100 | 68 / 100 | 17 / 100 |
| Llama-2-7b-hf | 0 / 0 | 0 / 5 | 0 / 0 | 0 / 34 | 0 / 22 | 0 / 0 |

Table 9: Well-learned digits / performance-preserving digits of models tested on NUPA Test according to dlength.

| | Add Int | Add Float | Add Frac | Add Easy Frac | Add Sci | Sub Int | Sub Float | Sub Frac |
|---|---|---|---|---|---|---|---|---|
| GPT-4o-mini | 20 / 20 | 4 / 8 | 0 / 2 | 1 / 2 | 0 / 0 | 20 / 20 | 5 / 9 | 0 / 1 |
| GPT-4o | 20 / 20 | 4 / 7 | 0 / 2 | 1 / 2 | 0 / 0 | 20 / 20 | 5 / 10 | 0 / 1 |
| Qwen2-72B | 20 / 20 | 7 / 13 | 0 / 0 | 0 / 0 | 0 / 0 | 20 / 20 | 0 / 0 | 0 / 0 |
| Qwen2-7B | 11 / 12 | 0 / 13 | 0 / 0 | 0 / 0 | 0 / 0 | 10 / 12 | 0 / 12 | 0 / 0 |
| Llama-3.1-8B-ft | 19 / 20 | 11 / 20 | 0 / 7 | 0 / 5 | 0 / 7 | 20 / 20 | 13 / 20 | 0 / 5 |
| Llama-3.1-70B | 20 / 20 | 6 / 13 | 0 / 1 | 0 / 1 | 0 / 0 | 20 / 20 | 4 / 13 | 0 / 1 |
| Llama-3.1-8B | 11 / 20 | 4 / 11 | 0 / 1 | 0 / 0 | 0 / 0 | 10 / 20 | 6 / 10 | 0 / 1 |
| Mixtral-8x7B | 10 / 14 | 4 / 12 | 0 / 1 | 0 / 1 | 0 / 0 | 16 / 20 | 5 / 13 | 0 / 0 |
| Llama-2-7b-hf | 5 / 12 | 0 / 10 | 0 / 0 | 0 / 0 | 0 / 0 | 4 / 20 | 0 / 11 | 0 / 1 |

| | Sub Sci | Max Int | Max Float | Max Frac | Max Sci | Max Hard Int | Max Hard Float | Max Hard Sci |
|---|---|---|---|---|---|---|---|---|
| GPT-4o-mini | 0 / 0 | 39 / 98 | 6 / 9 | 0 / 4 | 13 / 17 | 100 / 100 | 6 / 8 | 6 / 16 |
| GPT-4o | 0 / 0 | 31 / 72 | 6 / 9 | 0 / 2 | 12 / 17 | 100 / 100 | 6 / 8 | 8 / 16 |
| Qwen2-72B | 0 / 0 | 18 / 32 | 13 / 30 | 0 / 0 | 43 / 86 | 68 / 85 | 16 / 26 | 35 / 63 |
| Qwen2-7B | 0 / 0 | 11 / 23 | 0 / 17 | 0 / 0 | 0 / 14 | 86 / 100 | 0 / 20 | 0 / 11 |
| Llama-3.1-8B-ft | 0 / 0 | 47 / 83 | 35 / 54 | 0 / 5 | 97 / 100 | 100 / 100 | 25 / 52 | 35 / 69 |
| Llama-3.1-70B | 0 / 0 | 0 / 5 | 0 / 13 | 0 / 0 | 0 / 5 | 0 / 8 | 0 / 13 | 0 / 4 |
| Llama-3.1-8B | 0 / 0 | 0 / 0 | 0 / 7 | 0 / 0 | 0 / 0 | 0 / 18 | 0 / 6 | 0 / 0 |
| Mixtral-8x7B | 0 / 0 | 7 / 10 | 3 / 9 | 0 / 0 | 0 / 5 | 20 / 28 | 3 / 11 | 0 / 6 |
| Llama-2-7b-hf | 0 / 0 | 0 / 10 | 0 / 0 | 0 / 0 | 0 / 0 | 100 / 100 | 0 / 0 | 0 / 11 |

| | Multiply Hard Int | Multiply Hard Float | Multiply Hard Frac | Multiply Hard Sci | Multiply Easy Int | Multiply Easy Float | Multiply Easy Frac | Multiply Easy Sci |
|---|---|---|---|---|---|---|---|---|
| GPT-4o-mini | 7 / 11 | 0 / 0 | 1 / 2 | 0 / 0 | 9 / 20 | 0 / 0 | 1 / 2 | 0 / 0 |
| GPT-4o | 7 / 11 | 0 / 0 | 1 / 2 | 0 / 0 | 16 / 20 | 0 / 0 | 1 / 2 | 0 / 0 |
| Qwen2-72B | 0 / 6 | 0 / 0 | 0 / 0 | 0 / 0 | 5 / 7 | 0 / 0 | 0 / 0 | 0 / 0 |
| Qwen2-7B | 0 / 0 | 0 / 0 | 0 / 0 | 0 / 0 | 0 / 5 | 0 / 0 | 0 / 0 | 0 / 0 |
| Llama-3.1-8B-ft | 11 / 18 | 3 / 9 | 0 / 11 | 0 / 7 | 16 / 19 | 8 / 17 | 1 / 15 | 0 / 20 |
| Llama-3.1-70B | 5 / 9 | 0 / 3 | 0 / 1 | 0 / 0 | 7 / 16 | 0 / 3 | 0 / 1 | 0 / 0 |
| Llama-3.1-8B | 0 / 4 | 0 / 0 | 0 / 0 | 0 / 0 | 0 / 8 | 0 / 0 | 0 / 1 | 0 / 0 |
| Mixtral-8x7B | 4 / 6 | 0 / 0 | 0 / 2 | 0 / 0 | 4 / 8 | 0 / 0 | 0 / 2 | 0 / 0 |
| Llama-2-7b-hf | 0 / 0 | 0 / 0 | 0 / 0 | 0 / 0 | 0 / 5 | 0 / 0 | 0 / 0 | 0 / 0 |

| | Digit Max Int | Digit Max Float | Digit Add Int | Digit Add Float | Truediv Int | Truediv Frac | Floordiv Int | Mod Int |
|---|---|---|---|---|---|---|---|---|
| GPT-4o-mini | 5 / 16 | 0 / 7 | 0 / 8 | 0 / 0 | 0 / 5 | 1 / 2 | 8 / 14 | 3 / 15 |
| GPT-4o | 10 / 16 | 3 / 7 | 0 / 8 | 0 / 0 | 0 / 5 | 1 / 2 | 8 / 14 | 3 / 15 |
| Qwen2-72B | 7 / 16 | 0 / 9 | 0 / 0 | 0 / 0 | 0 / 0 | 0 / 1 | 5 / 7 | 0 / 5 |
| Qwen2-7B | 6 / 11 | 0 / 4 | 0 / 20 | 0 / 9 | 0 / 0 | 0 / 1 | 4 / 6 | 0 / 5 |
| Llama-3.1-8B-ft | 23 / 61 | 19 / 29 | 26 / 61 | 29 / 45 | 3 / 20 | 0 / 3 | 9 / 18 | 0 / 15 |
| Llama-3.1-70B | 6 / 16 | 0 / 7 | 0 / 11 | 0 / 0 | 0 / 0 | 0 / 1 | 3 / 10 | 0 / 4 |
| Llama-3.1-8B | 0 / 6 | 0 / 0 | 0 / 10 | 0 / 7 | 0 / 0 | 0 / 0 | 4 / 8 | 0 / 5 |
| Mixtral-8x7B | 0 / 12 | 0 / 4 | 0 / 8 | 0 / 7 | 0 / 0 | 0 / 1 | 4 / 7 | 0 / 5 |
| Llama-2-7b-hf | 0 / 7 | 0 / 3 | 0 / 9 | 0 / 7 | 0 / 0 | 0 / 1 | 0 / 3 | 0 / 6 |

| | Mod Easy Int | To Float Frac | To Float Sci | To Scient Int | To Scient Float | Sig Int |
|---|---|---|---|---|---|---|
| GPT-4o-mini | 3 / 20 | 0 / 4 | 0 / 15 | 9 / 21 | 5 / 8 | 0 / 23 |
| GPT-4o | 3 / 20 | 0 / 4 | 0 / 15 | 9 / 21 | 3 / 8 | 0 / 23 |
| Qwen2-72B | 0 / 10 | 3 / 8 | 12 / 46 | 0 / 44 | 0 / 0 | 6 / 19 |
| Qwen2-7B | 0 / 7 | 3 / 7 | 76 / 93 | 0 / 0 | 0 / 0 | 0 / 5 |
| Llama-3.1-8B-ft | 0 / 17 | 3 / 9 | 14 / 78 | 27 / 33 | 16 / 38 | 0 / 0 |
| Llama-3.1-70B | 0 / 5 | 3 / 7 | 0 / 15 | 0 / 0 | 0 / 0 | 0 / 0 |
| Llama-3.1-8B | 0 / 6 | 3 / 6 | 0 / 12 | 0 / 0 | 0 / 0 | 0 / 0 |
| Mixtral-8x7B | 0 / 6 | 3 / 6 | 15 / 16 | 0 / 18 | 0 / 0 | 0 / 0 |
| Llama-2-7b-hf | 0 / 6 | 0 / 4 | 93 / 98 | 0 / 0 | 0 / 0 | 0 / 0 |

Table 14: 8-digit digit match accuracy with small model (1.3M) with RoPE, NoPE and Alibi.

| | Int-add | Float-add | Fraction-multiplication | Scientific-add |
|---|---|---|---|---|
| RoPE | **0.091** | **0.88** | **0.23** | **0.75** |
| NoPE | 0.061 | 0.39 | 0.17 | 0.52 |
| Alibi | 0.056 | 0.31 | 0.18 | 0.50 |

Table 15: 8-digit digit match accuracy with small dataset (1M samples) with RoPE, NoPE and Alibi.

|  | Int-add | Float-add | Fraction-multiplication | Scientific-add |
|---|---|---|---|---|
| RoPE | **0.97** | **0.99** | **0.33** | **0.99** |
| NoPE | 0.78 | 0.98 | 0.29 | 0.96 |
| Alibi | 0.23 | 0.80 | 0.17 | 0.79 |

Table 17: Exact match of 0.1B models trained on integer addition, multiply and maximum respectively with various compositions of reverse formatting and index hints.

|  | Integer Addition | | | | Integer Multiply | | | | Integer Max | | | |
|---|---|---|---|---|---|---|---|---|---|---|---|---|
|  | rev | rev + idx | no | idx | rev | reverse + idx | no | idx | reverse only | reverse + idx | no | idx |
| d9 | **1.00** | 0.93 | 0.98 | 0.41 | **0.43** | 0.00 | 0.13 | 0.00 | **1.00** | 0.99 | **1.00** | 0.99 |
| d10 | **0.80** | 0.06 | 0.32 | 0.01 | **0.13** | 0.02 | 0.04 | 0.02 | **1.00** | 0.97 | **1.00** | 0.98 |

Table 18: Finetuning with PE, data format, and tokenizer modification will degrade the performance. The first two lines are a *naive finetuned* Llama and the original Llama *without finetuning*, which are the baseline. "1d" means using the *one-digit tokenizer* for numbers otherwise the *original tokenizer*. "rev" means *reverse* representation, where the integer parts are reversed. All the checkpoint we select by the lowest valid loss. The accuracy reported is the average "exact match" in each range. Metric "wld" is used to denote well-learned digit; "ppd" is used to denote performance-preserving digit.

|  | Integer Addition | | | | | | Float Addition | | | | | |
|---|---|---|---|---|---|---|---|---|---|---|---|---|
|  | S | M | L | XL | wld | ppd | S | M | L | XL | wld | ppd |
| FT | **0.95** | **0.65** | **0.12** | 0.01 | **4** | **12** | **0.96** | **0.71** | **0.27** | **0.08** | **5** | **17** |
| w/o FT | **0.95** | 0.38 | 0.06 | **0.02** | **4** | 9 | 0.90 | 0.47 | 0.10 | 0.02 | 3 | 11 |
| NoPE | 0.67 | 0.04 | 0.00 | 0.00 | 3 | 5 | 0.37 | 0.06 | 0.00 | 0.00 | 0 | 0 |
| NoPE + rev + 1d | 0.89 | 0.35 | 0.06 | **0.02** | 3 | 9 | 0.81 | 0.38 | 0.09 | 0.01 | 0 | 11 |
| NoPE + rev + pad + 1d | 0.87 | 0.34 | 0.05 | **0.02** | 0 | 9 | 0.74 | 0.38 | 0.06 | 0.01 | 0 | 9 |
| RoPE + 1d | 0.93 | 0.59 | 0.05 | 0.00 | 4 | 9 | 0.33 | 0.30 | 0.06 | 0.01 | 0 | 9 |
| RoPE + rev + 1d | 0.40 | 0.20 | 0.04 | 0.00 | 0 | 7 | 0.35 | 0.30 | 0.09 | 0.02 | 0 | 11 |

|  | Fraction Multiplication (easy) | | | | | | Scientific Notation Addition | | | | | |
|---|---|---|---|---|---|---|---|---|---|---|---|---|
|  | S | M | L | XL | wld | ppd | S | M | L | XL | wld | ppd |
| FT | **0.43** | 0.00 | 0.00 | 0.00 | **1** | 3 | **0.09** | **0.08** | **0.02** | 0.00 | 0 | **4** |
| w/o FT | 0.28 | 0.00 | 0.00 | 0.00 | 0 | 3 | 0.02 | 0.02 | 0.01 | 0.00 | 0 | 0 |
| NoPE | 0.14 | 0.00 | 0.00 | 0.00 | 0 | 3 | 0.00 | 0.00 | 0.00 | 0.00 | 0 | 0 |
| NoPE +rev + 1d | 0.25 | 0.00 | 0.00 | 0.00 | 0 | 3 | 0.01 | 0.02 | 0.01 | 0.00 | 0 | 0 |
| NoPE + rev + pad + 1d | 0.27 | 0.00 | 0.00 | 0.00 | 0 | 3 | 0.01 | 0.02 | 0.01 | 0.00 | 0 | 0 |
| RoPE + 1d | 0.09 | 0.00 | 0.00 | 0.00 | 0 | 1 | 0.08 | 0.03 | **0.02** | 0.00 | 0 | 3 |
| RoPE + rev + 1d | 0.06 | 0.00 | 0.00 | 0.00 | 0 | 1 | 0.08 | 0.02 | 0.01 | 0.00 | 0 | **4** |

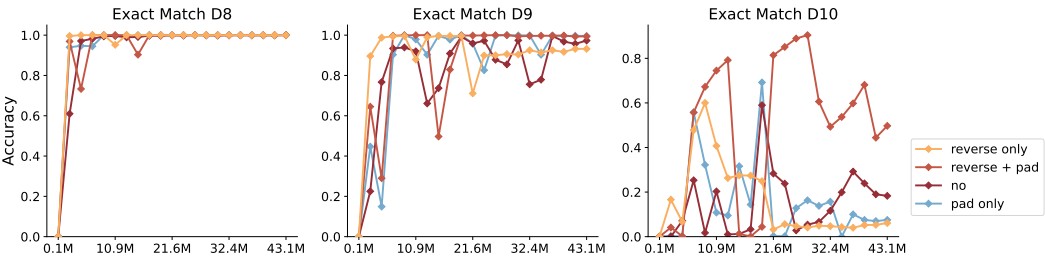

Figure 14: Exact match of 0.1B models trained on 1- to 8- digit integer addition with different compositions of reverse formatting and zero padding on 8- to 10- digit tests. X-axis is the number of seen training samples.

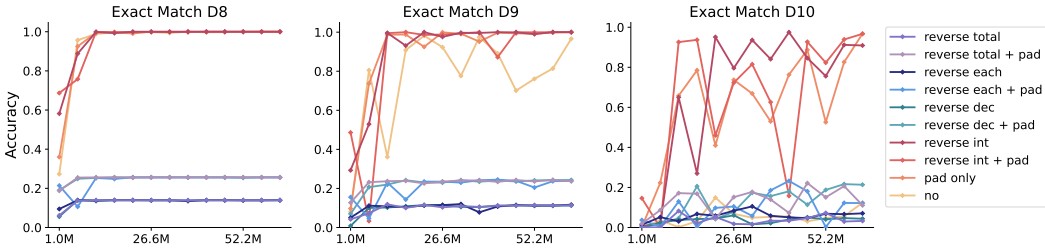

Figure 15: Exact match of 0.1B models trained on 1- to 8- digit float addition with different compositions of reverse formatting and zero padding on 8- to 10- digit tests. X-axis is the number of seen training samples.

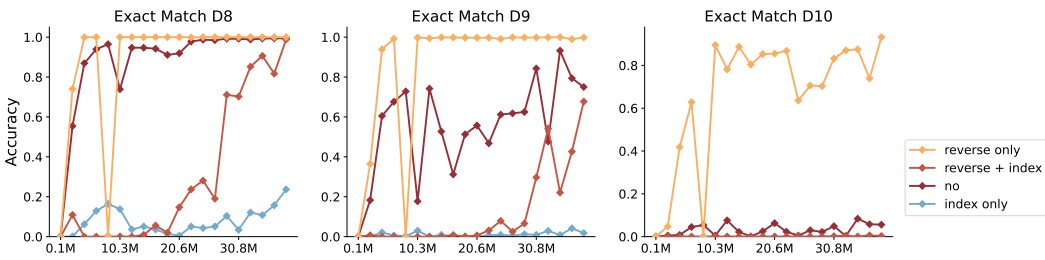

Figure 16: Exact match of 0.1B models trained on 1- to 8- digit integer addition with different compositions of reverse formatting and index hints on 8- to 10- digit tests. X-axis is the number of seen training samples.

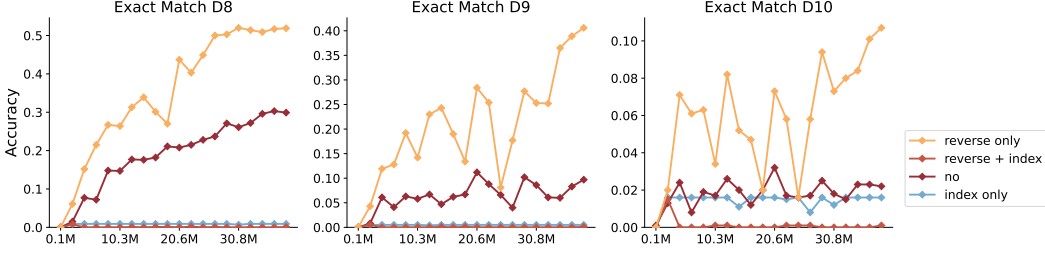

Figure 17: Exact match of 0.1B models trained on 1- to 8- digit integer multiplication with different compositions of reverse formatting and index hints on 8- to 10- digit tests. X-axis is the number of seen training samples.

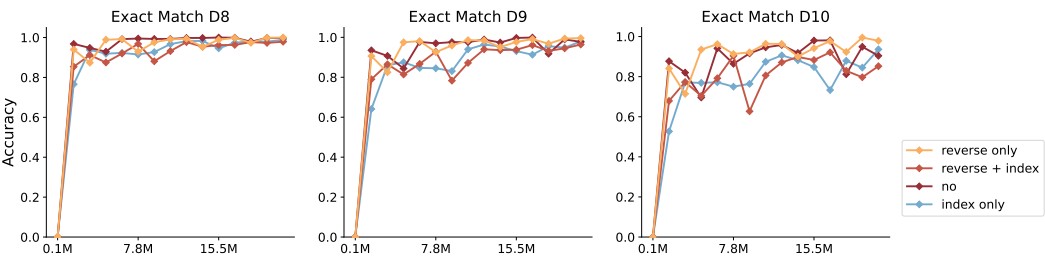

Figure 18: Exact match of 0.1B models trained on 1- to 8- digit integer maximum with different compositions of reverse formatting and index hints on 8- to 10- digit tests. X-axis is the number of seen training samples.

Table 19: Maximum length of each task that 2k context window can afford with RF-CoT

|  | Add | Sub | Multiply | Floordiv | Mod | Max | DigitMax | GetDigit | Length |
|---|---|---|---|---|---|---|---|---|---|
| Integer | 20 | 20 | 12 | 20 | 6 | 100 | 17 | 100 | 34 |
| Float | 6 | 5 | 4 | - | - | 50 | - | 100 | - |
| Fraction | 3 | 2 | 3 | - | - | 20 | - | - | - |
| Scientific | 3 | 3 | 3 | - | - | 100 | - | - | - |

