# OpenReview forum: "Number Cookbook: Number Understanding of Language Models and How to Improve It"
_ICLR.cc/2025/Conference — ICLR 2025 Poster_

### Official Review · Reviewer_Z2vn · 2024-10-16

**Soundness:** 2
**Presentation:** 2
**Contribution:** 2
**Rating:** 6
**Confidence:** 4

**Summary:**

This work introduces a benchmark suite of "numerical understanding and processing ability" (NUPA) tasks for transformer-based LLMs. The task suite separates numeracy from potentially confounding logical reasoning problems, and presents a variety of numerical formats. Additionally, various tricks are explored to improve LLM performance in these numeracy tasks; no silver bullets.

**Strengths:**

It is clear that some thought has gone into curating the benchmark tasks and justifying their inclusion. Even though I do not personally agree with some of the justifications and normative statements made, I believe that this sort of conscientious effort to justify a benchmark is commendable. I rate this work highly in terms of integrity, quality of presentation, and clarity.

I particularly appreciated the authors' investigation of CoT, as it is generally interesting to understand whether "prompt-level" interventions have a lot of room for improving performance on complex tasks. A recent (contemporaneous, so not factoring into my decisions) paper by Apple on GSM8k qualifying LLM performance as memorisation seems to agree with the findings of this work that numeracy abilities do not appear to scale; potentially a way to frame and situate this work with respect to the broader discourse on generalisation and understanding in LLMs.

**Weaknesses:**

I have no issues with the benchmark and evaluation. I do not currently find the conceptual motivations and wrapping compelling; I could be convinced otherwise for each of these with the help of citations, or of better arguments, and I would be happy to raise my score accordingly.

1. I am unconvinced that numeracy in LLMs is a problem in need of a solution. First, surely there is a citable source for LLM inadequacy for numeracy. Second, even if they were terrible at numeracy, the onus is on the authors to convince the reader that this a problem worth caring about, for at least two obvious reasons: 1) all of these tasks are already trivially done by a calculator or a python program, and 2) commercially available LLMs can probably do alright at numerical tasks indirectly via code-generation and execution. As it stands, it reads as if the authors are insisting that this is a problem deserving of attention --- I'm sure it could be, but this argument can be better made.

2. I am unconvinced that numeracy in LLMs is a problem in search of deeper understanding. Consider that "How many r's in strawberry" is a meme that normal people know about; the idea that tokenization impairs the ability of transformers to reason at a letter-, digit-, or word-unit level is well-digested. So when the authors claim that the weakness of LLMs at digit understanding is surprising (line 298), I find this a bit sensationalist and not grounded with respect to a broader and basic level of discourse around the capabilities of LLMs.

3. I am unconvinced of the normative rationales supporting the benchmarks, which seem ad-hoc. Here is an example: the authors claim that multiplication (not a particular algorithm) is O(L^2) in the length L counted in digits (of presumably both inputs; line 169), and take this as justification that models ought to find multiplication difficult. This doesn't hold up to armchair introspection: lookup is O(1), and we have all learned our times tables as children. The competent-human process of mental multiplication is mostly a sequence of lookups along with additive de/recomposition, so the real exponent is probably less than 2. This is a jarring oversight when contrasted with the effort with which the authors bring up cognitive and developmental psychology (lines 116 to 127), recasting what I assumed was a scholarly inclusion as a kind of checklisting exercise. There are other unsupported normative claims about what a good benchmark ought to encompass, such as "NUPAs are necessary and expected to learn by LLMs [sic]" (line 114), "any student who has completed primary and secondary education should be able to accomplish them" (line 160), and "for most practical tasks involving numbers (like arithmetic or math tests), all we care about is whether the answer is right, and there is no significant difference between being almost right and being completely wrong" (line 228). While I understand the importance of establishing normative standards to justify a benchmark with respect to, I find it hard to believe as a reader that the authors have the expertise and authority to do so.

4. I am unconvinced that this is worth "taking seriously"; let us suppose that everyone agrees that NUPA in LLMs is a problem worth solving and that improving on this benchmark is the way to solve the problem. We know beforehand that transformers have problems with NUPA due to tokenization, and now the finding of this paper is that fine-tuning and CoT as tricks don't solve the problem. There are many graphs in this paper that must have taken a fair amount of compute to obtain; is it worth it if everyone decided to run similar tests, or fine-tune with respect to the NUPA suite? I don't mean this in an unnecessarily antagonistic way, but one concerned with scientific interest and downstream impacts; if LLMs are just bad at numeracy for architectural or structural reasons as the current evidence suggests, could the introduction of a benchmark be inviting high-compute low-innovation approaches to the problem, and does the problem even merit this kind of resource investment?

**Questions:**

1. Why is it important that LLMs be good at NUPA when we python, and even LLM code-generation that can write python to deal with NUPA? I.e. Why is this a worthwhile problem, and who says it is?

2. To what extent do your findings suggest that the issue is architectural or to do with tokenization? To the degree that this is the case, to what extent do your findings suggest that the issue is paradigmatic rather than technical?

3. Is it necessary to provide normative justifications for the benchmarking suite? If so, I would like to know why, because I think the paper could be stronger if the benchmarking tests were presented matter-of-factly, without detours.

4. Experimentally, how much compute (as an estimate) was used throughout?

---

> ### Author Response · Authors · 2024-11-23
> **Official Rebuttal to Reviewer Z2vn (1/3)**
>
> We sincerely thank you for your thoughtful and constructive suggestions! We have revised the paper based on your feedback and hope the updates effectively address your concerns.
>
> 1. **Numeracy in LLMs is a problem**
>
>  - Although it is widely recognized that LLMs struggle with numeracy, there is still an absence of detailed benchmarks to evaluate specific challenges. Questions like which numerical tasks, number ranges, and task complexities are most difficult for models remain unanswered. A clear, detailed task definition is critical, rather than relying on vague notions of commonsense.
>  - We believe that NUPA without relying on external tools like Python is essential for an AGI candidate. Number processing is a high-frequency task, and dependence on such tools introduces significant overhead, increases complexity, and reduces parallelism [1]. Therefore, robust intrinsic numeracy is, in our view, critical for achieving efficient performance.
>  - Math ability is a key focus for LLM evaluation, with most models reporting metrics like MATH or GSM8k. Numerical processing is integral to this ability, and poor performance in NUPA directly hinders overall math capability. Notably, these metrics are reported without external tools, emphasizing the importance of intrinsic numeracy [2,3].
>  - While it is reasonable to use external tools for particularly complex problems, such as those involving very large or intricate numbers, models should not be expected to rely on tools for every numerical task. Therefore, it is crucial to establish a reference that identifies tasks the model can handle independently with high accuracy and those that necessitate external tool support. This distinction underscores the importance and value of a comprehensive benchmark.
>
> 2. **Tokenizer**
>
> - While it is commonly suggested that tokenization affects digit-level performance, its impact remains underexplored. Numbers differ fundamentally from text, as noted in the revised Section 3.1. Further research is needed to understand tokenization's role in handling numbers. As highlighted in our paper, while the trend favors larger vocabularies, one-digit tokenization performs best for NUPA. This finding is novel and has not been adequately addressed by open-source model trainers, warranting consideration for future tokenization design.
> - When we express surprise at LLMs’ struggle with digits, we mean that digits are foundational to arithmetic and math. If a model cannot reliably identify digits, its ability to solve complex math problems is questionable.
> - From an architectural perspective, the poor performance on digits might not be surprising. However, considering the model's expected mathematical capabilities (as many models, such as GPT-4 and Qen2.5, claim to possess strong mathematical abilities), it does seem unexpected. We have revised the paper to clarify this point, but if you still find it unclear or misleading, we’re more than happy to provide further clarification.

---

> ### Author Response · Authors · 2024-11-23
> **Official Rebuttal to Reviewer Z2vn (2/3)**
>
> 3. **Absolute and misleading statements**
>    We acknowledge that some statements in the earlier version of our paper may have appeared too absolute or misleading. These have been revised for greater accuracy and moderation in the updated submission. Examples include:
>
>    - "Any student who has completed primary and secondary education should be able to accomplish them" -> "Because these tasks are extracted from the education curricula, students who have completed the stage of education are expected to solve them."
>    - "for most practical tasks involving numbers (like arithmetic or math tests), all we care about is whether the answer is right, and there is no significant difference between being almost right and being completely wrong" -> "for most practical tasks involving numbers (like arithmetic or math tests), the correctness of the answer is the most important." In addition, this statement is a concession. We recognize and emphasize the importance of other metrics in just the next sentence "But having a smoother ...".
>    - We maintain that NUPAs are essential and should be expected to be learned by LLMs even when tools or code are available for assistance. As discussed earlier, relying on tools is not always an optimal solution for numerical processing. However, we have expanded and clarified our explanation in the updated version to make this point more reasonable and better aligned with practical considerations.
>    - The statement about $O(n^2)$ is not related to our conclusion and we are glad to remove it. However:
>      1. In context, we discuss how RF-CoT handles multiplication by following an algorithm with $O(n^2)$ complexity. While table lookups and additions may simplify this, they still break the problem into a double loop structure inherent to multiplication.
>      2. Even with shortcuts like a multiplication table, the complexity remains $O(n^2)$, as reducing constant terms (e.g., $O(n/2 * n/2) = O(n^2/4)=O(n^2)$) does not change the overall complexity.
>
>    However, we want to emphasize that the scope of tasks and their representations were chosen carefully, with reference to the Chinese primary and secondary school curricula, ensuring that the tasks are both common and representative.
>
> 4. **Concerning about high-compute low-innovation approaches**
>
> - While tokenization is an important factor, its effects on number processing remain poorly understood. There is no consensus on the best tokenization strategy, as current open-source models employ diverse approaches. Our work provides a valuable reference point for future model design. This area of research is still in its infancy, with many techniques proposed but their effectiveness in practical, diverse settings largely unproven. In this case,  providing a comprehensive and clear benchmark can contribute to more systematic and orderly progress in this field.
> - Although our paper includes extensive experiments, future researchers need not replicate all of them. For fine-tuning a model to enhance NUPA, only the fine-tuning experiments are required—there’s no need to re-run tokenization, PE, or data format experiments. Similarly, testing a model on NUPA only requires inference on our test set, which is computationally lightweight.
> - Our benchmark is computationally efficient. Training and inference focus on a small set of numbers, requiring minimal compute resources. For example, fine-tuning Llama on a single GPU takes approximately 5 hours, with inference requiring only 8 hours on the same setup. While some experiments from scratch (e.g., in Sections 3.1 and 3.2) are more demanding, even these remain manageable: 0.1B models take about 5 hours, 1B models about 20 hours, and 3B models about 40 hours on a single H800 GPU. Importantly, these more intensive experiments are not required for end users.

---

> ### Author Response · Authors · 2024-11-23
> **Official Rebuttal to Reviewer Z2vn (3/3)**
>
> 5. **Architecture or tokenization**
>
>    We believe it’s not necessary to prioritize either architecture or tokenizer exclusively. As our article demonstrates, both factors matter: architectural choices, such as position encoding (PE), are critical, and tokenizer behavior also impacts performance. Our aim is to provide guidance on both aspects rather than favoring one at the expense of the other.
>
>    Additionally, we suspect that limited data diversity during training contributes to the challenges observed. Notably, even simple fine-tuning, without additional techniques, significantly improves model performance.
>
>    In summary, we are confident that technical solutions exist, whether through better tokenizer and PE selection, or by developing new, more effective schemes. Our benchmark serves as a foundation for evaluating these approaches with clarity and comprehensiveness, supporting progress in this field toward more effective solutions.
>
> 6. **Rationale**
>
>    We believe it is essential to provide a clear rationale for the benchmark suite, given the diversity of number-related tasks and representations. When discussing model NUPA, the focus should be on tasks that are (1) genuinely important and common, and (2) appropriately challenging—neither too simple nor overly complex—to ensure the benchmark’s utility and relevance. Our justification aligns with these principles.
>
>    That said, we recognize that some statements in the original version may have been unclear or detracted from the main message. In the updated version, we have streamlined the content, keeping essential scenarios where these representations are relevant and omitting unnecessary details, such as how they are introduced in mathematics. We hope these revisions improve readability.
>
> Thank you again for your thoughtful feedback. If you have further suggestions, we are happy to consider them.
>
>
>
> [1] Xu et al.  Conveyor: Efficient Tool-aware LLM Serving with Tool Partial Execution, 2024
>
> [2] Meta, Introducing Llama 3.1: Our most capable models to date, 2024
>
> [3] Yang et al. Qwen2 Technical Report, 2024

---

> > ### Comment · Reviewer_Z2vn · 2024-11-23
> >
> > These are better arguments and citations, and my concerns feel addressed. **I will revise my overall score to 6.**, though I reserve the right to modulate again depending on discussion with the other reviewers. I believe that some of the other reviews may have been unnecessarily harsh, and **I intend to advocate on the authors' behalf, using their arguments, in discussion with other reviewers.**
> >
> > Nice complexity argument! Even if multiplication can be reduced to single-digit-times-table lookup, addition, and digit-shifts, you are right that it's the double loop that gives us quasipolynomial $\mathcal{O}(mn)$ in the two digit lengths, which becomes $\mathcal{O}(n^2)$ for $n$-by-$n$.

---

> > > ### Author Response · Authors · 2024-11-24
> > > **Official Comment by Authors**
> > >
> > > Thank you once again for your thoughtful review and recognition of our work. We hope this research will contribute to new insights and tools for understanding numerical concepts. We also welcome further discussions with you and would be glad to hear any additional suggestions you may have.

---

### Official Review · Reviewer_m8mv · 2024-10-27

**Soundness:** 3
**Presentation:** 2
**Contribution:** 3
**Rating:** 8
**Confidence:** 4

**Summary:**

Designs a comprehensive benchmark for number understanding, covering four different number presentations (e.g. integers) and 17 tasks (such as multiplication). The benchmarks allows for testing a range of different numerical understanding skills, at different levels of difficulty (through longer numbers in terms of digits for example). The authors do a comprehensive range of experiments on the benchmarks, evaluating off-the-shelf LLMs on it, as well as training their own models to understand the effects of aspects of common LLMs (such as tokenizers) and existing methods for improving numerical understanding (such as representing in reverse order). The findings are the many models still struggle in elementary understanding tasks, especially for larger numbers. Moreover, the existing tricks to deal with this don't fully mitigate the problems.

**Strengths:**

This paper has the potential to be an important contribution to the field, for two main reasons:

- Great benchmark proposed with a comprehensive range of tasks and difficulty levels, and good metrics for evaluating performance

- Excellent coverage of experiments. Testing comprehensively many aspects relevant to this problem, like different types of generalisation, different architectural changes, different existing models, effects of scale, existing methods for improving numeric understanding, prompting, etc. I would be very interested in these results, if the weaknesses below could be addressed.

**Weaknesses:**

Unfortunately, the presentation is lacking and it's very hard to interpret the results because of missing details on how models are trained.

**Main weaknesses**
- There is no information at all in the paper about how the models are trained for the experiments in section 3.1 and 3.2. How much examples do you train on? How long? Optimizers, hyperparameters, repeats, etc.? How do you define in and out-of-domain? It's very difficult to interpret the results of this experiment without knowing more.

- The paper needs substantial rewrites, and will probably benefit from corrections by someone whose first language is English or by using an LLM to revise and point out mistakes.

- There is a long discussion of things like the relevance of integers and how fractions arise (section 2.1 and 2.2), but this seems unnecessary and the space could be used instead to represent the results better. For example, Figure 2 presenting the main results in the text is very small and almost impossible to read without zooming in 200%. The colors used are also hard to distinguish. I would suggest shortening section 2.2 and almost entirely removing 2.1 and use the space to represent the results better. Especially because this is the great part of this paper, you have so many results but you do not discuss the details of the experiments and present the results in a way that are hard to follow.

- Some results are interpreted in ways that are not substantiated by the evidence found. For example, line 298 to 309; I disagree that this result means the model does not understand the concept of digit. The fact that it becomes harder to return the right digit for longer numbers actually hints at something else going on that might be more related to the model's inherent difficulties with retrieving something from a position in longer sequences, which has nothing to do with it's understanding of the concept digit. I also disagree that this points to case-based reasoning, because again it might be due to some aspects of its architecture. Additionally, the fact that models of different sizes show the same performance on a task does not necessarily indicate the performance depends on the training approach over size, it might also depend on architecture.

- There is no section on related work, and the authors say in the beginning of section 2 that they will show limitations of prior benchmarks while discussing the coverage of their own, but this does not happen. This makes it difficult for the reader to place this contribution w.r.t. existing literature.

**Minor points**

- Would be great to already get some more concrete information in the abstract (what probability of error? which 3 factors influence it?). Or at least in the intro some more concrete info.

- Figure 4 and 5: it's unclear what's on the X-axis (though one can assume it's training steps), and also unclear what D6 to D10 refers to. It's also unclear what is in-domain and what is OOD without reading the text.

- *"These types of errors are a major cause of hallucinations when dealing with math, reasoning, and data analysis tasks, as the model presents seemingly correct problem solving approaches but ultimately produces incorrect results"* -> this statement requires a citation

- line 117 to 119; a discussion on the innateness of integer understanding seems somewhat out of scope for this work and not too relevant to the contribution.

- Cite / reference openai chatgpt when using it line 135

- One approach that seems missing is few-shot examples, which might significantly boost performance for all models. Not just because they might learn from the examples, but because they get primed on the output format required.

**Questions:**

Main questions is can you give details on how the experiments that use training of fine-tuning are done, and what in-domain and out-of-domain refers to (what examples do you hold out for this)?

---

> ### Author Response · Authors · 2024-11-23
> **Official Rebuttal to Reviewer m8mv (1/2)**
>
> We sincerely thank you for your constructive feedback and valuable suggestions. We have revised the paper based on your suggestions and hope the updates address your concerns.
>
> 1. **Training details**
>    We train our models using an autoregressive Transformer architecture based on Llama-3.1 (unless stated otherwise). Sections 3.1 and 3.2 cover training from scratch with all hyperparameters, except model size, aligned with the original Llama setup. The AdamW optimizer is used with a learning rate of 5e-5, weight decay of 0.01, and batch sizes of 256, 64, and 32 for 0.1B, 0.9B, and 3B models, respectively, following default settings in the Transformers library.
>
>    Each model is trained on a consolidated dataset, comprising $10^7$ samples for each length (where feasible). Models are trained for one epoch using a cosine decay learning rate scheduler, and the best checkpoint on validation data is reported.
>
>    While experiments weren’t initially replicated at submission, the updated version includes three replicates for key experiments, reporting means and standard errors in figures and tables.
>
>    Our experiments were conducted on a cluster with Nvidia H800 GPUs (80GB memory). Training a 100M model from scratch takes 5–8 hours, a 1B model about 1 day, and a 3B model approximately 2 days on a single H800 GPU. Fine-tuning a pretrained model typically requires around 5 hours.
>
>    These details are included in the updated version (Appendix A.4.1) for improved reproducibility.
>
> 2. **In-domain and Out-of-domain**
>    The terms "in-domain" and "out-of-domain" refer to the **length** of numbers. As described in Section 3.1, the model is trained on numbers of lengths 1 to 8 (20) and tested on numbers of lengths 1 to 20 (100). Thus, lengths 1 to 8 (20) are considered in-domain, while lengths 9 (21) to 20 (100) are out-of-domain.
>
> 3. **Section 2.1 and 2.2**
>    The primary goal of our work is to propose a comprehensive benchmark to formalize NUPA. We believe it’s important to provide clear reasoning behind the choice of representations and tasks. To enhance clarity, we have significantly streamlined this section in the updated submission. In Section 2.1, we have retained the essential scenarios where these representations are relevant while omitting unnecessary details, such as their mathematical introductions. Likewise, Section 2.2 has been reorganized and condensed for improved coherence. We appreciate your suggestion to enhance the paper’s readability and invite you to review the updated version for further details.
>
> 4. **Figure 2**
>    Figure 2 has been updated for better readability. Additionally, to further enhance clarity, we have provided an interactive performance report as an anonymous *HTML* page [here](https://huggingface.co/spaces/NUPA-Anonymous/Performance). This allows readers to **interact with** the figure by selecting models, tasks, and metrics for a more detailed exploration.
>
> 5. **Interpretation about the results**
>    We recognize that some explanations in our initial submission may have led to misunderstandings due to less precise wording.
>
>  - For instance, when we state that "*the model does not understand the concept of digit*," we mean that the model cannot consistently solve digit-related tasks. Here, "concept" refers to a "set of digit-centered abilities" as defined in Section 2.2. We clarify that our paper acknowledges models can comprehend task instructions: "*models can at least comprehend the task instruction.*" This statement has been rephrased for clarity in the updated submission. Thank you for pointing this out.
>  - Regarding "*case-based reasoning*", we intended it as the opposite of rule-based reasoning, where the latter implies that models learn the **rules** of tasks to solve them consistently. (See Reference [1] for more details.) However, case-based reasoning does not contradict dependence on architecture. As noted in Section 3, architectures like RoPE might enable case-based reasoning, potentially offering shortcuts based on sequence length. Recognizing the term’s potential to mislead, we have removed it in the updated version.
>
>  - We agree that architecture plays a role and have added this to the listing. However, we maintain that the training approach is more critical. For instance, within the same architecture (e.g., LLaMA), fine-tuned versions with fewer parameters demonstrate significantly better performance than the pretrained versions, as highlighted in Section 3.

---

> ### Author Response · Authors · 2024-11-23
> **Official Rebuttal to Reviewer m8mv (2/2)**
>
> 6. **Related work**
>    Thank you for your suggestion. In the updated submission, we have added a related work section that now includes relevant datasets and benchmarks. To summarize, our work differs from previous benchmarks by treating NUPA as an independent task, distinct from math, language, or commonsense abilities. Additionally, we provide a comprehensive and detailed analysis of diverse numerical representations and tasks, which sets our approach apart.
>
> 7. **For these minor points:**
>
>    - The abstract has been revised and the vague terms have been removed.
>
>    - More explanations have been added in the caption of figures. Specifically, the X-axis is the seen samples, Dn is the number length (digit).
>
>    - The citation about the calculation hallucination has been added.
>
>    - The discussion on innateness has been removed.
>
>    - The citation of gpt-4o has been added. The original occurrence of "chatgpt" has been removed due to the space limitation.
>
>    - Regarding few-shot learning: In practical applications of number processing, such as financial reporting or solving arithmetic problems, we believe models should be capable of handling numbers without reliance on few-shot examples. Moreover, we view zero-shot reasoning ability as a critical direction for future model development.
>
>      Regarding the output format, we have reviewed the models’ outputs and found no issues, as our tasks require only simple numerical outputs. We have also tested some models with the results presented in Appendix A.3.1, and the conclusions remain consistent. Thank you for your valuable feedback!
>
> Additionally, we have thoroughly revised and polished the paper to improve its readability and clarity. We hope the updated version is now more accessible and understandable. We are happy to make further improvements if you have any additional suggestions. Thank you for your constructive feedback!
>
> [1] Hu et al. Case-Based or Rule-Based: How Do Transformers Do the Math?, 2024.

---

> ### Author Response · Authors · 2024-11-25
> **Looking Forward to Further Discussions**
>
> Dear Reviewer,
>
> As the discussion period is nearing its end in two days, we would like to follow up to ensure that our revisions and responses have addressed your concerns.
>
> In particular, we have made several updates to the paper.
> 1. We have added detailed information on the model and training process to provide greater clarity on our methodology. (Appendix A.4.1.)
> 2. Sections 2.1 and 2.2 have been streamlined to focus more on the relevance and importance of the content, removing less pertinent details.
> 3. We have improved the phrasing and clarity in the section on result interpretation. (Section 2.4)
> 4. We have added a section about related work. (Section 5)
>
> If you have any further questions or suggestions, we would be eager to continue the discussion with you over the next few days. Your feedback is highly valued, and we are keen to ensure that the manuscript fully meets your expectations.
>
> Thank you again for your thoughtful review.
>
> Best regards,
> Authors

---

> > ### Comment · Reviewer_m8mv · 2024-11-25
> > **Thanks for the comprehensive response**
> >
> > Dear authors,
> >
> > Thank you for the comprehensive response. My main weaknesses have all been addressed, so I will increase my support for this submission. I just wanted to briefly respond to two things in your response that I disagree with, but I leave it up to the authors to do anything with this or not.
> >
> > Firstly, claiming the model does not understand the concept of a digit because it cannot do tasks requiring returning a digit well. This can be explained by many different things besides the concept of a digit the model has. As your paper also highlights, there can be other things going on that does not allow the model to return a specific digit (like tokenizer problems), which can co-exist with the model having a good "concept of a digit". I respectfully disagree that the ability to consistently solve digit-related tasks directly probes the concept of a digit the model has. In any case, it seems like you already changed the wording around this topic in the submission.
> >
> > Secondly, even though case-based is the opposite of rule-based, a lack of rule-based behaviour does not imply case-based reasoning. It can imply imperfect applications of rule-based behaviour. Again, seems like you already changed this in the submission, but I just wanted to highlight this.
> >
> > Finally, although I agree that zero-shot improvements are important, it doesn't mean few-shot does not need to be tested. Even if a model fails zero-shot, it might very well just be because we are "using it wrong" (i.e. it just needs few-shot examples to properly respond). This is supported by results finding that things like fine-tuning stages on LLMs don't teach the model new capabilities, but enhance existing ones, and that techniques like best-of-N can sometimes be as good as fine-tuning; the capabilities are often already there in the model, we just need to properly use them to get them out. This is totally fair for a generalist model like an LLM, and often there is a simple few-shot prompt that improves performance across a lot of related tasks simply because it clarifies the requested output format. Again, zero-shot improvements are important, but if a model cannot do something zero-shot, few-shot capabilities should always be tested in a paper like this one that aims to map a models capabilities.
> >
> > I think this paper represents a strong submission now because it proposes a comprehensive benchmark and does a lot of experiments trying to understand when and why the numerical capabilities of LLMs are lacking.

---

> > > ### Author Response · Authors · 2024-11-26
> > > **Grateful for Your Invaluable Feedback**
> > >
> > > Thank you for your detailed and invaluable feedback. Your sightful suggestions have significantly improved the quality of our paper. We agree with your suggestions about "concept", "case-based learning" and "few-shot learning".
> > >
> > > 1. The use of term "concept" was indeed misleading. What we actually intended to highlight was that "the model fails to solve a series of tasks related to digits". we believe that the current revision will eliminate any misunderstandings.
> > >
> > > 2. For case-based learning, the reference [1] actually use more experimental evidence to support the case-based learning, which we do not specifically investigate in our experiments. We acknowledge that drawing conclusions about case-based learning at this stage may be premature, but we see this as a potential avenue for future research.
> > >
> > > 3. We agree with your perspective regarding the few-shot and we are now conducting further experiments to provide more conclusive results in final version.
> > >
> > > Once again, we deeply appreciate your meticulous and thought-provoking review. Your input has been invaluable to us, and we are grateful for the time and effort you have dedicated to our paper. We hope that this work contributes meaningfully to the ongoing developments in the field of LLM, and we are excited about the potential it holds for advancing the capabilities of LLMs.

---

> ### Author Response · Authors · 2024-12-03
> **About Few-shot Learning**
>
> Thank you for your suggestion and now we have finished our experiments with few-shot learning. Specifically, we test the open-source models with 5-shot examples in the same task and find that though in most cases, few-shot learning can generally improve performance, the conclusions in our paper are still satisfied.
>
> Our results are shown in [fig1](https://anonymous.4open.science/api/repo/NUPA_temp-3711/file/nupa_performance_exact_match_1.pdf?v=5f28507c) and [fig2](https://anonymous.4open.science/api/repo/NUPA_temp-3711/file/nupa_performance_exact_match_2.pdf?v=7bb49fc3). For example, the performance also significantly decreases as the length increases or the tasks and representations become unfamiliar (like Add-Fraction, Add-Scientific or floordiv). And the performance of digit-related tasks are still unsatisfying.
>
> Thank you for your reminder again. We believe it is an important supplementary and we will update them in the final version.

---

> > ### Comment · Reviewer_m8mv · 2024-12-03
> >
> > That's great! Thanks for your engagement with my suggestions

---

### Official Review · Reviewer_T1ZM · 2024-11-01

**Soundness:** 2
**Presentation:** 2
**Contribution:** 3
**Rating:** 6
**Confidence:** 4

**Summary:**

This paper evaluates language models on tasks involving numerical and arithmetic reasoning. The authors introduce a test suite with 17 numerical tasks (e.g., addition, subtraction, modulo, max, min, and digit retrieval) across four numerical representations, using it to evaluate nine language models. The paper also analyzes the impact of tokenization, positional embedding type, and data format on a model’s ability to perform addition and multiplication. Finally, the authors assess whether fine-tuning, with and without chain-of-thought, can improve LLaMA-3 8B’s performance.

**Strengths:**

- The evaluation procedure is well-structured and comprehensive.
- Some tasks in the evaluation suite offer useful insights into model failure modes.

**Weaknesses:**

- Attribution and discussion of previous work is extremely poor:
    - Notably, a discussion of the related work is missing. The the proposed evaluation suite, the design choices for it, and the results obtained should be discussed in light of previous studies that evaluated and analyzed the performance of language models on task involving numerical reasoning [e.g., 1,2,3,4,5,6]
    - The authors fail to reference papers that introduce methods that they directly adopt: chain-of-thought prompting was introduced by Wei et al. [7], and LoRA was proposed by Hu et al. [8]. Both citations are missing.
- Although the authors study the impact of tokenization on the performance of a model that they train form scratch, a discussion of how the tokenization of the pre-trained models evaluated might affect the results is missing. Additionally, here again, any reference to previous work that studied the impact of tokenization on numerical reasoning [e.g., 9] is absent.
- Some presentation/clarity issues:
    - Presenting the fine-tuning results in Table 2 is confusing, as they are discussed only much later in Section 4.
    - Figure 2 can be improved, especially in the choice of color for the LLaMA models, which are quite hard to distinguish in the bar plot.
    - Line 512.5: I understand that rule-following fine-tuning is introduced by Hu et al., but further details about the fine-tuning process (potentially in the appendix) would be helpful.
    - The use of negative \vspace is excessive, resulting in cramped spacing between tables/figures and text, especially on pages 7 and 9.

---
[1] A Survey of Deep Learning for Mathematical Reasoning (Lu et al., ACL 2023)
[2] Do NLP Models Know Numbers? Probing Numeracy in Embeddings (Wallace et al., EMNLP-IJCNLP 2019)
[3] Birds have four legs?! NumerSense: Probing Numerical Commonsense Knowledge of Pre-Trained Language Models (Lin et al., EMNLP 2020)
[4] A Causal Framework to Quantify the Robustness of Mathematical Reasoning with Language Models (Stolfo et al., ACL 2023)
[5] Injecting Numerical Reasoning Skills into Language Models (Geva et al., ACL 2020)
[6] Impact of Pretraining Term Frequencies on Few-Shot Numerical Reasoning (Razeghi et al., Findings 2022)
[7] Wei, Jason, et al. "Chain-of-thought prompting elicits reasoning in large language models." Advances in neural information processing systems 35 (2022): 24824-24837.
[8] Hu, Edward J., et al. "Lora: Low-rank adaptation of large language models." arXiv preprint arXiv:2106.09685 (2021).
[9] Singh, A.K. and Strouse, D.J., 2024. Tokenization counts: the impact of tokenization on arithmetic in frontier llms. arXiv preprint arXiv:2402.14903.

**Questions:**

The lack of discussion on previous work is a significant drawback. I would be willing to reconsider my score if this is addressed in the paper.

---

> ### Author Response · Authors · 2024-11-23
> **Official Rebuttal to Reviewer T1ZM**
>
> Thank you for your insightful review. We have revised the paper based on your suggestions, and we hope the updated version and our responses below address your concerns effectively.
>
> 1. **Related work**
>    We acknowledge that, due to space constraints, some discussions of relevant literature were omitted in the initial version. We recognize their importance and have added these discussions in a newly created section. Specifically, we have incorporated your suggested papers, along with others, into this section. Additionally, the paper [9] has been cited in the original section 3.1 and we have talked about the results. In fact, we adapt the setting of the paper right-to-left tokenization in our experiments (we also emphasize this point in updated submission). However, our findings indicate that the one-digit tokenizer performs best, while left-to-right and right-to-left tokenizers yield equivalent results.
>
> 2. **tokenizer**
>
>    1. Because the tokenization is bounded to the model, it is impossible to isolate the influence of tokenization in pre-trained models without introducing other confounding factors.
>    2. In the updated submission, we include a comparison of fine-tuning pre-trained models with modified tokenization in Section 3.3 (see Table 17: Row "RoPE+1d," which represents the fine-tuned Llama model with a one-digit tokenizer while keeping other settings unchanged). Similar to other fine-tuned models with modifications, the performance of the fine-tuned model with the one-digit tokenizer is worse than both the vanilla fine-tuned model and the original pre-trained model. This aligns with our conclusion: this kind of modification should be directly applied in the pretraining stage, instead of an ad-hoc during finetuning.
>
> 3. **presentation**
>
>    1. The colors in Figure 2 have been updated to enhance readability. The finetuned model is added in the figure for direct comparison with other models, as we believe presenting it in a separate figure would be less optimal. We can add an explanation at the caption. To further improve clarity, we have provided an interactive performance report as an anonymous *HTML* page [here](https://huggingface.co/spaces/NUPA-Anonymous/Performance), where the readers can **interact with** the figure and select the models, tasks and metrics.
>    2. A brief introduction to RFFT has been added at the beginning of Section 4 in the main paper, along with a detailed explanation and an example in Appendix A.5.1. Furthermore, all prompts and code are included in the supplementary materials for reference.
>    3. The paper has been reorganized to avoid the excessive use of negative vspace, improving its overall structure and readability.
>
> Thank you once again for your detailed and thoughtful feedback.

---

> ### Author Response · Authors · 2024-11-25
> **Looking Forward to Further Discussions**
>
> Dear Reviewer,
>
> As the discussion period is set to conclude in two days, we would like to kindly follow up to ensure that our responses have addressed your concerns.
>
> Specifically, we have **revised and refined the presentation** further, with careful attention to **adding detailed content regarding related work**. The references you provided, along with other relevant paper, have now been thoroughly added into the paper.
>
> If you have any additional questions or suggestions, we would be delighted to engage in further discussions with you over the next few days. Your feedback is invaluable to us, and we are eager to ensure that the paper meets the highest standards.
>
> Best regards,
> Authors

---

> > ### Comment · Reviewer_T1ZM · 2024-11-25
> >
> > Thank you for your response and for the revisions to the paper. The discussion of related works contextualizes the contribution and the additional analyses included make the paper a stronger submission. I also appreciated the interactive HTML visualization of the results that you provided. I will raise my score to reflect these improvements.

---

> > > ### Author Response · Authors · 2024-11-26
> > > **Thank You for Your Thoughtful Review and Suggestions**
> > >
> > > Thank you again for your thoughftul review of our paper. Your constructive, especially the comprehensive reference, have been incredibly help us to improve our work. We hope this work can contribute to new insights and tools for understanding numerical concepts. Thank you again for your time and effort in reviewing our submission.

---

### Official Review · Reviewer_MwX6 · 2024-11-03

**Soundness:** 3
**Presentation:** 2
**Contribution:** 2
**Rating:** 6
**Confidence:** 3

**Summary:**

This paper investigates the empirical abilities of LLMs to solve numerical reasoning tasks of varying complexity. It proposes a benchmark, called NUPA, incorporating several different number representations (integers, fractions, etc) and reasoning tasks (arithmetic, comparisons, etc). Its experiments show that several well-known LLMs are prone to errors on this benchmark, particularly as the digits get larger. The paper also analyzes how factors like tokenization, data formats, and finetuning affect performance on NUPA.

**Strengths:**

1. The paper does a good job of broadening the evaluation of numerical reasoning from integers—which is often the focus in these kinds of studies—to more general classes of numbers. In general, there are a lot of experiments here, including several tasks and language models. The claim that models struggle with basic numerical understanding is well supported by the results.
2. The discussion on tokenization is quite interesting, surprisingly few studies that I am aware of consider how tokenization affects numerical reasoning. Although not very surprising, it is nice to have empirical evidence that 1-digit tokenization yields stronger generalization to more complex examples than those in the training set.
3. The paper goes beyond being purely “descriptive”, it also investigates methods that could *improve* numerical reasoning.

**Weaknesses:**

1. Numerical reasoning in LLMs is well-studied by now. Yet, the paper lacks a related work section and has very few references to similar studies overall. It is therefore unclear to what extent the insights here are novel. Some of the claims of novelty also come off as overly strong, e.g., the last sentence of the abstract “our work takes an initial step towards understanding and improving NUPA [numerical understanding and processing ability] of LLMs.” I include a short list of relevant references below; however, I would suggest the authors to perform a more comprehensive literature review so that they can properly situate their work.
2. The paper is somewhat poorly written. First, there are many grammatical errors in this text (e.g., l62-3, l123). Second, many parts lack explanation or justification. For instance, it is not explained how the random tokenizer works. The technique appears to be adapted from Sathe et al. (2024), but that is a recent and probably not very well-known paper. On another note, random tokenizers were not introduced by that paper as suggested in the text; see for instance Kudo (2018). Another part that is unclear to me is the section on positional encodings. It is not explained what is meant by length-agnostic and length-related rules. It also doesn’t contextualize the methods studied—RoPE and Alibi. What are they and why are they studied? Is there some plausible explanation for why you observe the results you do?
3. The paper is experiment-driven; there is no underlying theory guiding the selection of tasks or techniques for improvement. The scope is also very broad—the paper attempts to do many things at once. While that may be seen as a strength, I feel that it comes at the sacrifice of depth of analysis and clarity. I would suggest the authors to construct a more focused narrative and provide justifications that are grounded either in previous work or theory.
4. The paper lacks important statistical reporting like confidence intervals or p-values.
5. As a minor additional note, Figure 2 (and the similar figures in appendix) is quite hard to parse. There is a lot of information and it is difficult to distinguish the models since some of the colors are rather similar. It is also confusing that the results for the finetuned model are here since finetuning is discussed much later in the paper.


----

Dziri et al. 2023. Faith and Fate: Limits of Transformers on Compositionality.

Kudo et al. 2023. Do Deep Neural Networks Capture Compositionality in Arithmetic Reasoning?

Razeghi et al. 2022. Impact of Pretraining Term Frequencies on Few-Shot Numerical Reasoning.

Zhang et al. 2024. Interpreting and Improving Large Language Models in Arithmetic Calculation.

**Questions:**

How do you get ground truth reasoning traces for RFFT?

---

> ### Author Response · Authors · 2024-11-23
> **Official Rebuttal to Reviewer MwX6**
>
> Thank you for your valuable review. We have revise the paper according to your suggestions and we hope the updated version and the following answers can address your concerns.
>
> 1. **Related work**
>
>    Thank you for pointing out these related works. A related work section has been included in the revision now as section 5, which we hope will help readers better understand our contributions and novelty. We have included two of your suggested papers in the reference list and we find the other two papers about compositionality appear to be less relevant to our study.  If you have further suggestions or feedback, we would be happy to consider them.
>
>    We are also pleased to summarize the novelty of our work as follows:
>
>     (1) Unlike previous benchmarks, where NUPA is intertwined with math, language, or commonsense abilities, our benchmark isolates NUPA as an independent focus, allowing for a targeted and detailed analysis.
>
>     (2) While prior studies have highlighted the diversity of numerical representations and tasks, they often lack structured organization and comprehensive analysis of these aspects. In contrast, our work meticulously categorizes and analyzes numerical representations and tasks, offering a clear and thorough specification of their scope.
>
> 2. **overly strong novelty claim**
> Thank you for your reminder. When we  refer to our work as an initial step, we mean we first emphasize NUPA itself as an independent task, characterized by representation complexity and diversity, separate from math or reading comprehension. We acknowledge that some statements in the previous version may be unclear and overstated. We have revised them into more accurate ones like "*Our work provides a more detailed and comprehensive understanding of NUPA in LLMs*" in the updated submission.
>
> 3. **Add explanations**
> We include the following explanation in the updated version:
>
>  - For the **random tokenizer**, we add a brief introduction at the beginning of the paragraph "random tokenizer" as follows:
>       >Introduced as ``sub-word regularization'' by [5,6], the random tokenizer splits words like "Hello world" into variable tokens such as "He/llo/ world" or "Hell/o/ world". Though not widely used in LLMs, [7] found that it enhances reasoning by introducing variability in generation path. Inspired by this, we apply this to the numbers, segmenting numbers into tokens with lengths randomly chosen between 1 and a predefined maximum, instead of using greedy left-to-right segmentation.
>
>  - We clarify the distinction between "*length-related*" and "*length-agnostic*" at the beginning of section 3.2 (Now in appendix A.4.3 due to the space limitation). Using integer addition as an example, its rules are length-agnostic since addition involves processing numbers digit by digit, unaffected by their length. However, during training, because the model is exposed to numbers within a limited length range (e.g., 1 to 8), it may develop a length-related rule that combines the original addition rules with an artificial constraint like "the output length must be between 1 and 8."
>
>  - We add an *explanation of the PE* results at the end of Appendix A.4.3. In summary, we believe that RoPE allows models to learn positional and length information more effectively, which in turn encourages the model to adopt a length-related rule as a shortcut.
>  The question of "*why RoPE facilitates learning length information*" is intriguing but beyond the scope of this study. This phenomenon is likely tied to the architecture of the PEs. For instance, RoPE encodes positional information using a d-dimensional vector, whereas Alibi relies on a scalar, and NoPE uses no positional encoding at all. While we provide some intuition, a detailed analysis of the mechanisms behind PEs is a substantial topic on its own, and we look forward to further research in this direction.
>
>  - We add a explanation about what are and why we choose RoPEs, NoPEs and Alibi.
>       >RoPE, widely used in Llama and its derivatives, is the most classic relative PE. Then alibi, another relative PE, is proposed to address RoPE's length overfitting issues. NoPE (transformers without PE, relying solely on the causal mask to encode the position information) offers a surprisingly easy way to achieve length generalization. Therefore, we compare these three typical PEs to evaluate the performance on NUPA.

---

> > ### Author Response · Authors · 2024-11-23
> > **Official Rebuttal to Reviewer MwX6 (2/2)**
> >
> > 4. **Guidance for the selection of tasks and techniques**:
> > - The selection of tasks in our paper is carefully designed based on two criteria: (1) the task should be common and representative, which is why we reference (Chinese) primary and secondary school curricula and evaluate each candidate task for inclusion (2) the tasks should be of appropriate difficulty—not too easy or too hard—for the models to solve. We have added a detailed explanation of our task selection process. If you have suggestions for additional suitable tasks, we would be happy to consider and include them.
> > - Regarding the selection of techniques, we focused on those commonly used in numerical reasoning and length generalization research. We chose the most typical and representative techniques (tokenizer, PE and data format) to evaluate the performance of NUPA. We mainly aim to check the efficiency of these techniques on our newly proposed tasks. To make the paper readable, we have *moved some discussion to the Appendix*. We hope this revision can address your concern. If you have any further suggestions, please let us know and we are glad to further polish the paper.
> >
> > 5. **Statistical reporting**:
> > We repeat our mainly experiments three times and report the standard error (Figures & Tables) in the updated version.
> >
> > 6. **Figure 2**
> > We have updated the color scheme in Figure 2 to enhance readability. The finetuned model is added in the figure for direct comparison with other models, as we believe presenting it in a separate figure would be less optimal. We can add an explanation at the caption.
> > To further improve clarity, we have provided an interactive performance report as an anonymous *HTML* page [here](https://huggingface.co/spaces/NUPA-Anonymous/Performance), where the readers can **interact with** the figure and select the models, tasks and metrics.
> >
> > [1] Eric Wallace, et al.  Do NLP Models Know Numbers? Probing Numeracy in Embeddings, 2019.
> >
> > [2] Devin Johnson, et al.  Probing for multilingual numberical understanding in transformer-based language models, 2020.
> >
> > [3] Bill Yuchen Lin, et al.  Birds have four legs?! NumerSense: Probing Numerical Commonsense Knowledge of Pre-Trained Language Models, 2020.
> >
> > [4] Mubashara Akhtar, et al. Exploring the numerical reasoning capabilities of language models: A comprehensive
> > analysis on tabular data, 2023.
> >
> > [5] Taku Kudo.  Subword Regularization: Improving Neural Network Translation Models with Multiple Subword Candidates, 2018.
> >
> > [6] Ivan Provilkov, Dmitrii Emelianenko, and Elena Voita.  BPE-Dropout: Simple and Effective Subword Regularization, 2020.
> >
> > [7] Ashutosh Sathe, Divyanshu Aggarwal, and Sunayana Sitaram.  Improving self consistency in llms through probabilistic tokenization, 2024.

---

> > > ### Comment · Reviewer_MwX6 · 2024-11-24
> > >
> > > Thank you for the thorough clarifications and updates. I am raising my score.
> > >
> > > And great that you added standard errors to your plots and tables. They seem to still be missing in figure 2, however. For the future (and perhaps for that figure) I would recommend calculating error bars based on bootstrap sampling, eliminating the need to rerun experiments. https://en.wikipedia.org/wiki/Bootstrapping_(statistics)

---

> > > > ### Author Response · Authors · 2024-11-25
> > > > **Thank You for Your Thoughtful Review and Suggestions**
> > > >
> > > > Thank your again for your thorough and insightful review of our work. Your constructive comments and detailed feedback have been invaluable in helping us improve our paper. Your recommendation to use the bootstrap method is appreciated, we will add an analysis based on bootstrap sampling in final version. And we will further refine the figure sizes to improve clarity and include this interactive leaderboard in the main text to make it more accessible for readers.
> > > >
> > > > We truly value your expertise and input. If you have any further suggestions or additional feedback, we would be more than happy to consider them as we continue to improve our work.

---

### Author Response · Authors · 2024-12-02
**Acknowledgments and Summary of Revisions Based on Reviewer Feedback**

We greatly appreciate the reviewers for their careful review and valuable suggestions, which have significantly improved our paper. We have carefully considered all feedback, particularly the recommendations on related work, and incorporated them into the updated PDF. We are really grateful for the constructive discussions, which not only enhance our paper but also reflect a valuable and positive review process.

In summary, our paper focuses on **number understanding and processing abilities** (NUPA) of LLM, independent from other mathematical and reasoning tasks. We provide a comprehensive analysis of tasks involving various numerical representations and abilities, and propose a benchmark. Our tests show that while LLMs perform well on common tasks, their NUPA ability declines with less common numerical representations, tasks or longer sequences. We further explore pretraining techniques (tokenizers, PEs and data formats), fine-tuning, and CoT on NUPA tasks, concluding that the NUPA problem remains unsolved and future work should focus on improving number encoding and increasing task diversity.

We revised the paper based on the reviewers' suggestions: (1) reduced off-topic discussions when introducing representations and tasks; (2) added a related work section for clearer background; (3) refined conclusions for better accuracy; and (4) adjusted figure colors for readability. We also introduced an interactive website for better understanding of the experimental results. We thank the reviewers again for their efforts and are pleased that these revisions are successful to address their concerns. We hope our work will contribute to LLM research, particularly in handling numerical tasks more effectively.

---

### Meta-Review · Area_Chair_7RMb · 2024-12-19

**Metareview:**

The authors present a new benchmark for numerical reasoning and several experiments studying the effect of various manipulations on benchmark performance. This addresses an important problem, broadens the scope of previous benchmarks, and raises insights about model failures -- overall presents a clear and useful new contribution.

**Additional Comments On Reviewer Discussion:**

Clarity and contextualization issues resolved significantly. Overall strong engagement from reviewers and authors.

---

### Decision · Program_Chairs · 2025-01-22

Accept (Poster)